# Swing-by Dynamics in Concept Learning and Compositional Generalization

**Yongyi Yang**[1,2,3], **Core Francisco Park**[1,2,4], **Ekdeep Singh Lubana**[1,2], **Maya Okawa**[1,2],
**Wei Hu**[3], **Hidenori Tanaka**[1,2]

[1] CBS-NTT Physics of Intelligence Program, Harvard University, Cambridge, MA, USA
[2] Physics & Informatics Laboratories, NTT Research, Inc., Sunnyvale, CA, USA
[3] Computer Science and Engineering, University of Michigan, Ann Arbor, MI, USA
[4] Department of Physics, Harvard University, Cambridge, MA, USA

## ABSTRACT

Prior work has shown that text-conditioned diffusion models can learn to identify and manipulate primitive concepts underlying a compositional data-generating process, enabling generalization to entirely novel, out-of-distribution compositions. Beyond performance evaluations, these studies develop a rich empirical phenomenology of learning dynamics, showing that models generalize sequentially, respecting the compositional hierarchy of the data-generating process. Moreover, concept-centric structures within the data significantly influence a model's speed of learning the ability to manipulate a concept. In this paper, we aim to better characterize these empirical results from a theoretical standpoint. Specifically, we propose an abstraction of prior work's compositional generalization problem by introducing a structured identity mapping (SIM) task, where a model is trained to learn the identity mapping on a Gaussian mixture with structurally organized centroids. We mathematically analyze the learning dynamics of neural networks trained on this SIM task and show that, despite its simplicity, SIM's learning dynamics capture and help explain key empirical observations on compositional generalization with diffusion models identified in prior work. Our theory also offers several new insights—e.g., we find a novel mechanism for non-monotonic learning dynamics of test loss in early phases of training. We validate our new predictions by training a text-conditioned diffusion model, bridging our simplified framework and complex generative models. Overall, this work establishes the SIM task as a meaningful theoretical abstraction of concept learning dynamics in modern generative models.

## 1 INTRODUCTION

Human cognitive abilities have been argued to generalize to unseen scenarios through the identification and systematic composition of primitive concepts that constitute the natural world (e.g., shape, size, color) (Fodor, 2001; Fodor et al., 1975; Reverberi et al., 2012; Frankland & Greene, 2020; Russin et al., 2024; Franklin & Frank, 2018; Goodman et al., 2008). Motivated by this perspective, the ability to compositionally generalize to entirely unseen, out-of-distribution problems has been deemed a desirable property for machine learning systems, leading to decades of research on the topic (Smolensky, 1990; Lake & Baroni, 2018; Hupkes et al., 2020; Ramesh et al., 2021; Rombach et al., 2022; Lake & Baroni, 2023; Kaur et al., 2024; Du & Kaelbling, 2024).

Recent work has shown that modern neural network training pipelines can lead to emergent abilities that allow a model to compositionally generalize when it is trained on a data-generating process that itself is compositional in nature (Lake & Baroni, 2023; Ramesh et al., 2023; Okawa et al., 2023; Lepori et al., 2023; Arora & Goyal, 2023; Zhou et al., 2023; Khona et al., 2024; Rosenfeld et al., 2020; Nagarajan et al., 2020; Arjovsky et al., 2019). For example, Okawa et al. (2023); Park et al. (2024) show that text-conditioned diffusion models can learn to identify concepts that constitute the training data and develop abilities to manipulate these concepts flexibly, enabling generations that represent novel compositions entirely unseen during training. These papers also provide a spectrum of intriguing empirical results regarding a model's learning dynamics in a compositional task. For example, they reveal that abilities to manipulate individual concepts are learned in a sequential

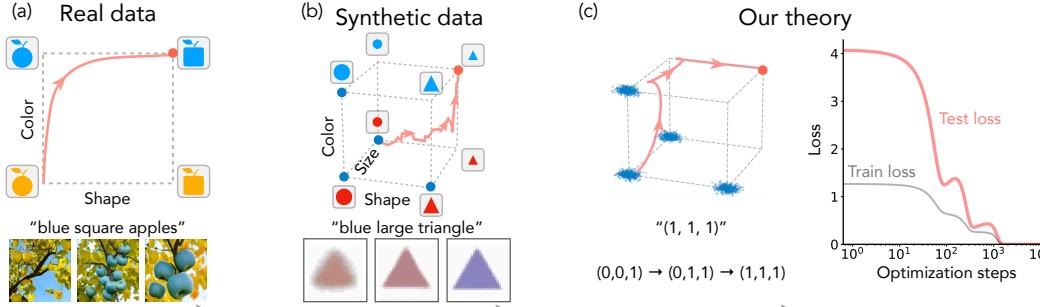

Figure 1: **Structured Identity Mapping Task and Swing-by Generalization Dynamics.** (a) Given the input "blue square apples on a tree with circular yellow leaves," a multimodal model learns to generate concepts in the following order: "apple," "blue" (color), and "square" (shape) (example adapted from Li et al. (2024)). (b) A multimodal synthetic task introduced by Okawa et al. (2023); Park et al. (2024). The training set of the task consists of four distinct compositions of concepts, depicted as blue nodes on a cubic graph. A diffusion model is trained on this dataset to systematically study the dynamics of concept learning. With the test prompt "small, blue, triangle," the diffusion model sequentially learns the correct size, shape, and finally color. (c) In this work, we introduce a structured identity mapping task as the foundation for a systematical and theoretical studying of the dynamics of concept learning. The model is trained on a Gaussian mixture data, where the centroids are positioned at certain nodes of a hyperrectangle (blue dots) and is evaluated on an out-of-distribution test set (red dot). Our theoretical results not only reproduce and explain previously characterized empirical phenomena but also depict a comprehensive picture of the non-monotonic learning dynamics in the concept space and predict a "multiple-descent" curve of the test loss (red curve).

order dictated by the data-generating process; the speed of learning such abilities is modulated by data-centric measures (e.g., gradient of loss with respect to concept values, such as color of an object); and the most similar composition seen during training often controls performance on unseen compositions.

In this work, we aim to demystify the phenomenology of compositional generalization identified in prior work and better ground the problem (or at least a specific variant of it called systematicity) via a precise theoretical analysis. To that end, we instantiate a simplified version of the compositional generalization framework introduced by Okawa et al. (2023); Park et al. (2024)—called the "concept space" (see Fig. 1b)—that is amenable to theoretical analysis. In brief, a concept space is a vector space that serves as an abstraction of real concepts. For each concept (e.g., color), a binary number can be used to represent its value (e.g., $0$ for red and $1$ for blue). In this way, a binary string can be mapped to a tuple (e.g., $(1, 0, 1)$ might represent "big blue triangle") and then fed into the diffusion model as a conditioning vector. The model output is then passed through a classifier[1] which produces a vector indicating how accurately the corresponding concepts are generated (e.g. a generated image of big blue triangle might be classified as $(0.8, 0.1, 0.9)$). In this way, the process of generation becomes a vector mapping, and *a good generator essentially performs as an identity mapping in the concept space*.

We argue that in fact the salient characteristic of a concept space is its preemptively defined organization of concepts in a systematic manner, not the precise concepts used for instantiating the framework itself. Grounded in this argument, we define a learning problem called the *Structured Identity Mapping (SIM) task* wherein a regression model is trained to learn the identity mapping from points sampled from a mixture of Gaussians with structurally organized centroids (see Fig. 1c). Through a detailed analysis of the learning dynamics of MLP models, both empirically and theoretically, we find that SIM, despite its simplicity, can both capture the phenomenology identified by prior work and provide precise explanations for it. Our theoretical findings also lead to novel insights, e.g., predicting the existence of a novel mechanism for non-monotonic learning curves (similar to epoch-wise double descent (Nakkiran et al., 2021), but for *out-of-distribution data*) in the early phase of training, which we empirically verify to be true by training a text-conditioned diffusion model. Our contributions are summarized below.

---

[1]Conceptually, we can think of an idealized perfect classifier here.

- **Structured Identity Mapping (SIM): A faithful abstraction of concept space.** We empirically validate our SIM task by training Multi-Layer Perceptrons (MLPs), demonstrating the reproduction of key compositional generalization phenomena characterized in recent diffusion model studies (Okawa et al., 2023; Park et al., 2024). Our findings show: (i) learning dynamics of OOD test loss respect the compositional hierarchical structure of the data generating process; (ii) the rate at which a model disentangles a concept and learns the capability to manipulate it is dictated by the sensitivity of the data-generating process to changes in values of said concept (called "concept signal" in prior work); and (iii) network outputs corresponding to weak concept signals exhibit slowing down in concept space. These results also suggest that the structured nature of the data, rather than specific concepts, drove observations reported in prior work.

- **Swing-by Dynamics: Theoretical analysis reveals mechanisms underlying learning dynamics of a compositional task.** Building on the successful reproduction of phenomenology with MLPs trained on the SIM task, we further simplify the architecture to enable theoretical analysis. We demonstrate that: (i) analytical solutions with a linear regression model reproduce the observed phenomenology above, and (ii) the analysis of a symmetric 2-layer network ($f(\boldsymbol{x}; \boldsymbol{U}) = \boldsymbol{U}\boldsymbol{U}^\top \boldsymbol{x}$) identifies a novel mechanism of non-monotonic learning dynamics in generalization loss, which we term **Swing-by Dynamics**. Strikingly, we show that the learning dynamics of compositional generalization loss can exhibit *multiple descents* in its early phase of learning, corresponding to multiple phase transitions in the learning process.

- **Empirical confirmation of the predicted Swing-by phenomenon in diffusion models.** We verify the predicted mechanism of Swing-by Dynamics in text-conditioned diffusion models, observing the non-monotonic evolution of generalization accuracy for unseen combinations of concepts, as predicted by our theory.

In summary, our theoretical analysis of networks trained on SIM tasks provides mechanistic explanations for previously observed phenomenology in empirical works and introduces the novel concept of Swing-by Dynamics. This mechanism is subsequently confirmed in text-conditioned diffusion models, bridging theory and practice in compositional generalization dynamics.

## 2 PRELIMINARIES AND PROBLEM SETTING

Throughout the paper, we use bold lowercase letters (e.g., $\boldsymbol{x}$) to represent vectors, and use bold uppercase letters (e.g., $\boldsymbol{A}$) to represent matrices. We use the unbold and lowercase version of corresponding letters with subscripts to represent corresponding entries of the vectors or matrices, e.g., $x_i$ represent the $i$-th entry of $\boldsymbol{x}$ and $a_{i,j}$ represent the $(i,j)$-th entry of $\boldsymbol{A}$. For a vector $\boldsymbol{x}$ and a natural number $k$, we use $\boldsymbol{x}_{:k}$ to represent the $k$-dimensional vector that contains the first $k$ entries of $\boldsymbol{x}$. For a natural number $k$, we use $[k]$ to represent the set $\{1, 2, \ldots, k\}$, and $\mathbf{1}_k$ to represent a vector whose entries are all 0 except the $k$-th entry being 1; the dimensionality of this vector is determined by the context if not specified. In the theory part, we frequently consider functions of time, denoted by variable $t$. If a function $g(t)$ is a function of time $t$, we denote the derivative of $g$ with respect to $t$ by $\dot{g}(t_0) = \frac{\mathrm{d}g}{\mathrm{d}t}\big|_{t=t_0}$. Moreover, we sometimes omit the argument $t$, i.e., $g$ means $g(t)$ for a time $t$ determined by the context. For a statement $\phi$, we define $\mathbb{1}_{\{\phi\}} = \begin{cases} 1 & \phi \text{ is true} \\ 0 & \phi \text{ is false} \end{cases}$ to be the indicator function of that statement.

### 2.1 PROBLEM SETTING

Now we formally define SIM, which is an abstraction of the concept space. For each concept class, we model them as a Gaussian cluster in the Euclidean space, placed along a unique coordinate direction. The distance between the cluster mean and the origin represents the strength of the concept signal, and the covariance of the Gaussian cluster represents the data diversity within the corresponding concept class. Additionally, we allow more coordinate directions than the clusters, meaning that some coordinate directions will not be occupied by a cluster, which we call non-informative directions, and they correspond to the free variables in the generalization task. See Fig. 1c for an illustration of the dataset of SIM.

**Training Set.** Let $d \in \mathbb{N}$ be the dimensionality of the input space and $s \in [d]$ be the number of concept classes, i.e., there are $s$ Gaussian clusters, and $n \in \mathbb{N}$ number of samples from each cluster. The training set $\mathcal{D} = \bigcup_{p \in [s]} \left\{ \boldsymbol{x}_k^{(p)} \right\}_{k=1}^n$ is generated by the following process: for each

$p \in [s]$, each training point of the $p$-th cluster is sampled i.i.d. from a Gaussian distribution $x_k^{(p)} \sim \mathcal{N}\left[\mu_p \mathbf{1}_p, \mathrm{diag}\,(\boldsymbol{\sigma})^2\right]$, where $\mu_p \geq 0$ is the distance of the $p$-th cluster center from the origin, and $\boldsymbol{\sigma}$ is a vector with only the first $s$ entries being non-zero, and $\sigma_i^2$ describing the data variance on the $i$-th direction. There is also optionally a cluster centered at $\mathbf{0}$ in addition to the $s$ clusters.

**Loss function.** The training problem is to learn identity mapping on $\mathbb{R}^d$. For a model $f : \mathbb{R}^m \times \mathbb{R}^d \to \mathbb{R}^d$ and a parameter vector $\boldsymbol{\theta} \in \mathbb{R}^m$, we train the model parameters $\boldsymbol{\theta}$ via the mean square error loss.

$$\mathcal{L}(\boldsymbol{\theta}) = \frac{1}{2sn} \sum_{p=1}^{s} \sum_{k=1}^{n} \left\| f\left(\boldsymbol{\theta}; x_k^{(s)}\right) - x_k^{(s)} \right\|^2. \tag{2.1}$$

**Evaluation.** We evaluate the model at a Gaussian cluster centered at the point that combines the cluster means of all training clusters. When the variance of the test set is small, the expected loss within the test cluster is approximately equivalent to the loss at its cluster mean. Therefore, for simplicity, in this paper, we focus on the loss at the sum of the test cluster, which is a single test point $\hat{x} = \sum_{p=1}^{s} \mu_p \mathbf{1}_p$. In App. B, we report further results for the case of various combinations of training clusters, which leads to multiple OOD test points.

## 3 OBSERVATIONS ON THE SIM TASK

We first begin by summarizing our key empirical findings on the SIM task. In all experiments we use MLP models of various configurations, including different number of layers and both linear and non-linear (specifically, ReLU) activations. Throughout this section and the subsequent sections, we frequently consider the model output at the test point $\hat{x}$ over training time, which we call **output trajectory** of the model.

Due to space constraints, we only present the results for a subset of configurations in the main paper and defer other results to App. F. We note that the findings reported in this section are in one-to-one correspondence with results identified using diffusion models in Sec. 5 and prior work (Park et al., 2024).

### 3.1 GENERALIZATION ORDER CONTROLLED BY SIGNAL STRENGTH AND DIVERSITY

One interesting finding from previous work is that if we alter the strength of one concept signal from small to large, the contour of the learning dynamics would dramatically change (Park et al., 2024). Moreover, it is also commonly hypothesised that with more diverse data, the model generalizes better (Gong et al., 2019; Díez-Pastor et al., 2015). Recall that in the SIM task, the distance $\mu_k$ of each cluster represents the corresponding signal strength, and the variance $\sigma_k$ represents the data diversity. In Fig. 2, we present the output trajectory under the setting of $s = 2$, in which case the trajectory can be visualized in a plane. There are two components to be learned in this task and, from the contour of the curve, we can tell the order of different components being learned.

Fig. 2 (a) presents the output trajectory for a setting with a fixed and balanced $\boldsymbol{\sigma}$, and a varied $\boldsymbol{\mu}$. The results show that when $\mu_1 < \mu_2$, the dynamics exhibit an upward bulging, indicating a preference for the direction of stronger signal. As $\mu_1$ is gradually increased, this contour shifts from an upward bulging to a downward concaving, and consistently maintains the stronger signal preference.

In Fig. 2 (b), the $\boldsymbol{\mu}$ is fixed to an unbalanced position, with one signal stronger than the other. As we mentioned above, when $\boldsymbol{\sigma}$ is balanced, the model will first move towards the cluster with a stronger signal strength. However, when the level of diversity of the cluster with weaker signal is gradually increased, the preference of the model shifts from one cluster to another.

A very concrete conclusion can be thus drawn from the results in Fig. 2 (a) and (b): the generalization order is jointly controlled by the signal strength and data diversity, and, generally speaking, the model prefers direction that has a stronger signal and more diverse data. We note that the conclusion here is more qualitative and in Sec. 4, we provide a more precise quantitative characterization of how these two values control the generalization order.

### 3.2 CONVERGENCE RATE SLOW DOWN IN TERMINAL PHASE

In Fig. 2, the arrow-like markers on the line indicate equal training time intervals. In the later phase of training, we observe that the arrows get denser, indicating a slowing down of the learning dynamics: at the terminal phase of training, the time required to reduce the number of training steps to reduce the same amount of loss is significantly larger than at the beginning.

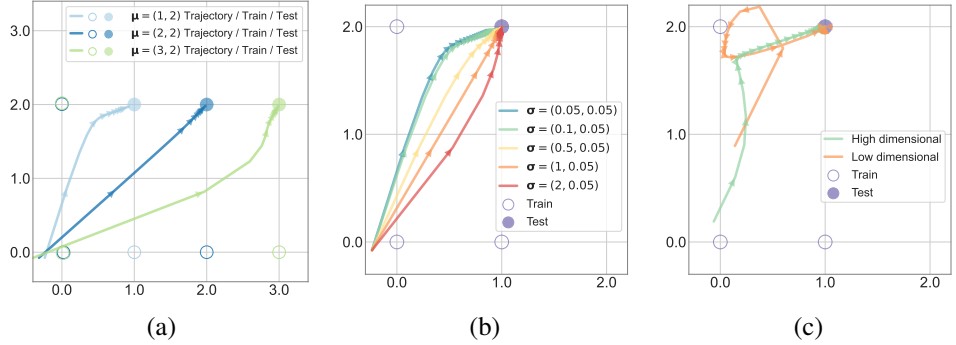

(a)              (b)              (c)

Figure 2: **Learning dynamics of MLP on SIM task.** The figures show the output trajectory of the MLP on a two-dimensional setting (i.e., $s = 2$), and each marker represents an optimization timepoint. Notice that we only plot the center of the training set as a circle, but the actual training set can have varied shapes based on the configuration of $\boldsymbol{\sigma}$. (a) one-layer linear model with $\boldsymbol{\sigma}_{:2} = (.05, .05)$ and varied $\boldsymbol{\mu}$. Concepts $i$ with larger signal ($\mu_i$) learnt first. (b) one-layer linear model with $\boldsymbol{\mu}_{:2} = (1, 2)$ and varied $\boldsymbol{\sigma}$. Concepts $i$ with larger diversity ($\sigma_i$) learnt first. (c) 4 layer linear models under $\boldsymbol{\mu}_{:2} = (1, 2)$ and $\boldsymbol{\sigma}_{:2} = (.05, .05)$ and different dimensionality. high dim: $d = 64$, low dim: $d = 2$. Notice that (a) and (b) are both in high dim setting. The lower the dimensionality, the stronger Swing-by it has.

### 3.3 SWING-BY DYNAMICS

The results in Fig. 2 (a) and (b) are both performed with one-layer models and under a high dimensional setting ($d = 64$). Despite the overall trend being similar in other settings, it is worth exploring the change of trajectory as we increase the number of layers, and / or reduce the dimension.

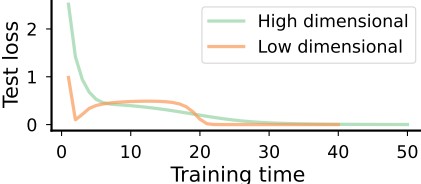

Figure 3: The test loss of multi-layer models.

In Fig. 2 (c), we perform experiments with deeper models, and optionally with a lower dimension. Under these changes, we find that the model shows an interesting irregular behavior, where it initially heads towards the OOD test point, but soon turns toward the training set cluster with the strongest signal. This indicates the model, while seems to be generalizing OOD at the beginning, is memorizing the train distribution and unable to generalize OOD at this point. However, with enough training, we find the model start to again move towards the test point and thus generalizes OOD. We call this overall dynamic of the output trajectory **Swing-by**, which we could be suggestive of a non-monotonic test loss curve. To assess this further, we track the value of the loss function during training in Fig. 3, demonstrating a double descent-like curve[2]. We also note that the *Swing-by* phenomena seems to be strongest when dimensionality $d$ of the dataset is low, and is rather modest with high dimensional settings. In the high dimensional setting, the OOD loss descent slows down at some point but does not actually exhibit non-monotonic behavior. This low dimensional preference can also be explained perfectly by our theory, further described in Sec. 4.

### 4 THEORETICAL EXPLANATION

We next study the training dynamics of a specific class of linear models that are tractable on the SIM task and explain the empirical phenomenology of OOD learning dynamics seen in previous section. In Sec. 4.1, we first analyze a one-layer model whose dynamics can be solved analytically. We show that it can explain most phenomena observed in the experiment; however, it fails to reproduce Swing-

---

[2]We would like to note that, it is possible to understand Swing-by Dynamics as a special case of so-called epoch-wise double descent (Nakkiran et al., 2021; Olmin & Lindsten, 2024; Schaeffer et al., 2023); however, epoch-wise double descent is generally understood as a consequence of either noisy training or over-parametrization, affecting model's in-distribution generalization. In contrast, Swing-by is a distributional phenomenon where the model fits the training *distribution* and is hence unable to generalize well OOD, revealing a novel mechanism that although leads towards a familiar double descent-like curve.

by Dynamics, suggesting that Swing-by Dynamics is intrinsic to deep models, which highlights the fundamental difference between shallow and deep models. In Sec. 4.2, we further analyze the dynamics of a symmetric 2-layer linear model, which successfully captures Swing-by Dynamics. Our theoretical results reveal a multi-stage behavior of the model Jacobian during training, which leads to the non-monotonic behavior in model output. We show that each stage in the Swing-by Dynamics precisely corresponds to each stage in the Jacobian evolution.

Throughout this section, we assume $\boldsymbol{f}(\boldsymbol{\theta}; \boldsymbol{x})$ is a linear function of $\boldsymbol{x}$. In this case the Jacobian of $\boldsymbol{f}$ with respect to $\boldsymbol{x}$ is a matrix that is completely determined by $\boldsymbol{\theta}$, which we denote by $\boldsymbol{W_\theta} = \frac{\partial f(\boldsymbol{\theta};\boldsymbol{x})}{\partial \boldsymbol{x}}$. In this way, the output of the model can be written as $\boldsymbol{f}(\boldsymbol{\theta}; \boldsymbol{x}) = \boldsymbol{W_\theta}\boldsymbol{x}$. Using the trace trick (with detailed calculations provided in App. C.1), it is easy to show that the overall loss function is equal to

$$\mathcal{L}(\boldsymbol{\theta}) = \frac{1}{2} \left\| (\boldsymbol{W_\theta} - \boldsymbol{I}) \, \boldsymbol{A}^{1/2} \right\|_{\mathcal{F}}^2, \tag{4.1}$$

where $\boldsymbol{A} = \frac{1}{sn} \sum_{p=1}^s \sum_{k=1}^n \boldsymbol{x}_k^{(p)} \boldsymbol{x}_k^{(p)\top}$ is the empirical covariance. In this section, we assume $n$ is large, in which case $\boldsymbol{A}$ converges to the true covariance of the dataset $\boldsymbol{A} = \mathbb{E}_{\boldsymbol{x}\sim\mathcal{D}}[\boldsymbol{x}\boldsymbol{x}^\top]$, which is a diagonal matrix $\boldsymbol{A} = \mathrm{diag}(\boldsymbol{a})$, defined by $a_p = \begin{cases} \sigma_p^2 + \frac{\mu_p^2}{s} & p \le s \\ 0 & p > s \end{cases}$, for any $p \in [d]$.

**Remark.** Notice that in the linear setting we might not directly train $\boldsymbol{W_\theta}$; instead, we train its components. For example, we might have $\boldsymbol{\theta} = (\boldsymbol{W}_1, \boldsymbol{W}_2)$ and have $\boldsymbol{W_\theta} = \boldsymbol{W}_1 \boldsymbol{W}_2$. Then, what we actually train is $\boldsymbol{W}_1$ and $\boldsymbol{W}_2$, instead of $\boldsymbol{W_\theta}$. As many previous works have emphasized (Arora et al., 2018; Ji & Telgarsky, 2018; Arora et al., 2019; Advani et al., 2020), although the deep linear model has the same capacity as a one-layer linear model, their dynamics can be vastly different and the loss landscape of deep linear models can be non-convex.

## 4.1 A One-Layer Model Theory and Its Limitations

As a warm-up, we first study the dynamics of one-layer linear models, i.e., $f(\boldsymbol{W}; \boldsymbol{x}) = \boldsymbol{W}\boldsymbol{x}$, in which case the Jacobian $\boldsymbol{W_\theta}$ is simply $\boldsymbol{W}$. As we will show, this setting can already explain most of the observed phenomenology from the previous section including the order of generalization and the terminal phase slowing down, but fails to capture the Swing-by Dynamics, which we will explore in next subsection. Here we present Theorem 4.1, which gives the analytical solution of the one-layer model on the SIM task.

**Theorem 4.1.** *Let $\boldsymbol{W}(t) \in \mathbb{R}^{d \times d}$ be initialized as $\boldsymbol{W}(0) = \boldsymbol{W}^{(0)}$, and updated by $\dot{\boldsymbol{W}} = -\nabla\mathcal{L}(W)$, with $\mathcal{L}$ be defined by eq. (4.1) with $f(\boldsymbol{W}, \boldsymbol{z}) = \boldsymbol{W}\boldsymbol{z}$, then we have for any $\boldsymbol{z} \in \mathbb{R}^d$,*

$$f(\boldsymbol{W}(t), \boldsymbol{z})_k = \underbrace{\mathbb{1}_{\{k \le s\}} \left[1 - \exp\left(-a_k t\right)\right] z_k}_{\widetilde{G}_k(t)} + \underbrace{\sum_{i=1}^s \exp\left(-a_i t\right) w_{k,i}(0) z_i}_{\widetilde{N}_k(t)}. \tag{4.2}$$

See App. C.2 for proof of Theorem 4.1. The Theorem shows that the $k$-th dimension of the output of a one-layer model evaluated on the test point $\hat{\boldsymbol{x}}$ can be decomposed into two terms: the *growth term* $\widetilde{G}_k(t) = \mathbb{1}_{\{k \le s\}} \left[1 - \exp\left(-a_k t\right)\right] \mu_k$, and the *noise term* $\widetilde{N}_k(t) = \sum_{i=1}^s \exp\left(-a_i t\right) w_{k,i}(0) \mu_i$. The following properties can be observed for these two terms: (i) the growth term converges to $\mu_k$ when $k \le s$ and 0 when $k > s$, while the noise term converges to 0; (ii) both terms converge at an exponential rate; and (iii) the noise term is upper bounded by $\sum_{i=1}^s w_{k,i}(0)\mu_i$. If the model initialization is small in scale, specifically $w_{k,i}(0) \ll \frac{1}{s \max_{i \in [s]} \mu_i}$, then $\widetilde{N}_k(t)$ will always be small, and thus can be omitted. With this assumption in effect, the model output is dominated by the growth term. A closer look at the growth term then explains part of the observed phenomenology.

**Generalization Order and Terminal Phase Slowing Down.** It can be observed that $\widetilde{G}_k(t)$ converges at an exponential rate, which leads an exponential decay of evolution speed and *explains the terminal phase slowing down*. Moreover, the exponential convergence rate of $\widetilde{G}_k(t)$ is controlled by the coefficient $a_k = \frac{1}{s} \left(s\sigma_k^2 + \mu_k^2\right)$. Therefore, the direction with larger $a_k$, i.e., larger $\mu_k$ and / or $\sigma_k$, converges faster, *hence explaining the order of generalization to different concepts*. The

theorem also reveals the proportional relationship between $\mu_k$ (concept signal strength) and $\sigma_k$ (data diversity).

**The Limitation of the One Layer Model Theory.** While we have demonstrated that Theorem 4.1 effectively explains both the generalization order and the terminal phase slowing down, in the solution eq. (4.2), the learning of each direction is independent. This independence omits the possible interaction between the dynamics of different directions in deeper models, and leads to monotonic and rather regular output trajectory (this is verified by the experiment results in Sec. 3.1). However, as the experiments in Sec. 3.3 show, when the number of layers becomes larger, the model actually exhibits a non-monotonic trace that can have detours. The theory based on the one-layer model fails in capturing this behavior. In the subsequent subsection, we introduce a more comprehensive theory based on a deeper model, and demonstrate that this model explains all the phenomena observed in Sec. 3, especially the Swing-by Dynamics.

## 4.2 A SYMMETRIC TWO-LAYER LINEAR MODEL THEORY

In this subsection, we analyze a symmetric 2-layer linear model, namely $f(\boldsymbol{U}; \boldsymbol{x}) = \boldsymbol{U}\boldsymbol{U}^\top \boldsymbol{x}$, where $\boldsymbol{U} \in \mathbb{R}^{d \times d'}$ and $d' \geq d$. We demonstrate that it accurately captures all the observations presented in Sec. 3, and, more importantly, the theory derived from this model provides a comprehensive understanding of the evolution of the model Jacobian and output, offering a clear and intuitive explanation for the underlying mechanism of the model's seemingly irregular behaviors. Due to space constraints, we focus on providing an intuitive explanation of the multi-stage behavior of the model Jacobian and output, and defer the formal proofs to the appendix. It is also worth noting that this symmetric 2-layer linear model is a frequently studied model in theoretical analysis (Li et al., 2020; Stöger & Soltanolkotabi, 2021; Jin et al., 2023), and most existing theoretical results for this model focus on the implicit bias of the solution found, instead of on the non-monotonic behavior during training, which is the focus of our analysis.

For convenience, we denote the Jacobian of $f$ at time point $t$ by $\boldsymbol{W}(t) = \boldsymbol{W}_{\boldsymbol{U}(t)}$. The gradient flow update of the $i, j$-th entry of $\boldsymbol{W}$ is given by

$$\dot{w}_{i,j} = \underbrace{w_{i,j}(a_i + a_j)}_{G_{i,j}(t)} - \underbrace{\frac{1}{2}w_{i,j}\left[w_{i,i}(3a_i + a_j) + \mathbb{1}_{\{i \neq j\}}w_{j,j}(3a_j + a_i)\right]}_{S_{i,j}(t)} - \underbrace{\frac{1}{2}\sum_{\substack{k \neq i \\ k \neq j}} w_{k,i}w_{k,j}(a_i + a_j + 2a_k)}_{N_{i,j}(t)}.$$

$$(4.3)$$

As noted in eq. (4.3), we decompose the update of $w_{i,j}$ into three terms. We call $G_{i,j}(t) = w_{i,j}(t)(a_i + a_j)$ the *growth term*, $S_{i,j}(t) = \frac{1}{2}w_{i,j}\left[w_{i,i}(3a_i + a_j) + \mathbb{1}_{\{i \neq j\}}w_{j,j}(3a_j + a_i)\right]$ the *suppression term*, and $N_{i,j}(t) = \frac{1}{2}\sum_{\substack{k \neq i \\ k \neq j}} w_{k,i}(t)w_{k,j}(t)(a_i + a_j + 2a_k)$ the *noise term*. The name of these terms suggests their role in the evolution of the Jacobian: the growth term $G_{i,j}$ always has the same sign as $w_{i,j}$, and has a positive contribution to the update, so it always leads to the direction that **increases the absolute value** of $w_{i,j}$; the suppression term $S_{i,j}$ also has the same sign[3] as $w_{i,j}$, but has a negative contribution in the update of $w_{i,j}$, so it always leads to the direction that **decreases the absolute value** of $w_{i,j}$; and the effect direction of the noise term is rather arbitrary since it depends on the sign of $w_{i,j}$ and other terms. It is proved in Lemma D.8 that under mild assumptions, the noise term will never be too large; for brevity, we omit it in the following discussion and defer the formal treatment of it to the rigorous proofs in App. D.

### 4.2.1 THE EVOLUTION OF ENTRIES OF JACOBIAN

In order to better present the evolution of the Jacobian, we divide the entries of the Jacobian into three types: the **major entries** are the first $s$ diagonal entries, and the **minor entries** are the off-diagonal entries who are in the first $s$ rows or first $s$ columns, and other entries are **irrelevant entries**. Notice that the irrelevant entries do not contribute to the output of the test point so we will not discuss them. Moreover, we also divide minor entries into several groups. The minor entries in the $p$-th row or column belongs to the $p$-th group (thus each entry belongs to two groups). See Fig. 4 for an illustration of the division of the entries.

---

[3]Notice that since $\boldsymbol{W} = \boldsymbol{U}\boldsymbol{U}^\top$ is a PSD matrix, the diagonal entries are always non-negative.

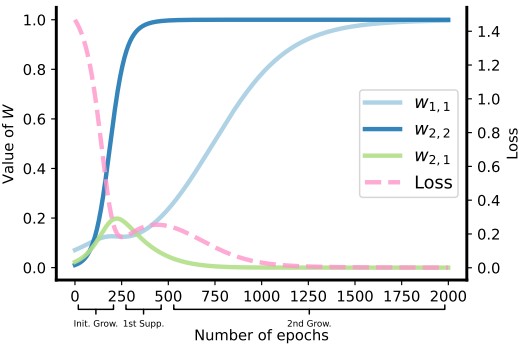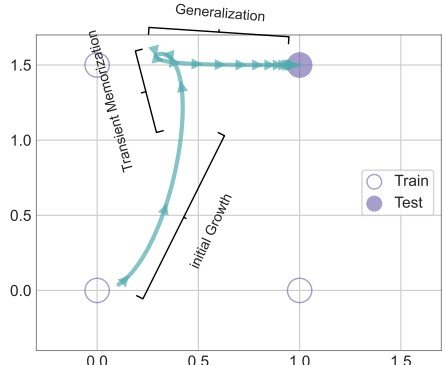

Figure 5: **The learning dynamics of a symmetric 2-layer linear model.** Left: The change of the test loss and the Jacobian entries with time predicted by the theory; Right: the corresponding model output trajectory. The figures are plotted under $s = 2$ and all entries of $W$ are initialized positive.

**Initial Growth.** In this section we assume $w_{i,j} \forall i, j$ are initialized around a very small value $\omega$ such that $\omega \ll \frac{1}{d \max_{i \in [s]} a_i}$ (This has been shown crucial for generalization (Xu et al., 2019; Liu et al., 2022b), and especially in compositional tasks (Zhang et al.); See App. D.1 for specific assumptions). It is evident that when all $w_{i,j}$ are close to $\omega$ (we call this period the **initial phase**), the growth term is $\Theta(\omega)$, while the suppression term and the noise term are $\Theta(\omega^2)$. This suggests that the evolution of $w_{i,j}$ is dominated by the growth term. Therefore, in the initial phase, every value in the Jacobian grows towards the direction of increasing its absolute value, with the speed determined by $a_i + a_j$. Since we assumed that $\boldsymbol{a}$ is ordered in a descending order, it is evident that each entry grows faster than those below it or to its right. The Initial Growth stage is formally characterized by Lemmas D.1 to D.3.

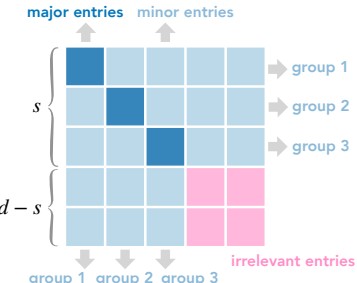

Figure 4: An illustration of the entries of the Jacobian.

**First Suppression.** In the Initial Growth stage, the first major entry will be the one that grows exponentially faster than all other entries, making it the first one that leaves the initial phase. Once the first major entry becomes significant and non-negligible, it will effect on the suppression term of all minor entries in the first group. When the difference between $a_1$ and $a_2$ is large enough, the first major entry is able to flip the growth direction of the first group of minor entries and push their values to 0. The suppression stages are characterized by Lemma D.7.

**Second Growth and Cycle.** Once the suppression of the first group of minor entries takes effect, the second major entry becomes the one that grows fastest. Thus, the second major entry will be the second one that leaves the initial stage. Again, when the second major entry becomes large enough, it will suppress the second group of minor entries and push their value to 0. This process continues like this: the growth of a major entry is followed by the suppression of the corresponding group of minor entries, which, in turn, leaves space for the growth of the next major entry. The general growth stages are characterized by Lemma D.4 and the fate of off-diagonal entries is characterized by Lemma D.8.

**Growth Slow Down and Stop.** Notice that the suppression term of a major entry is also influenced by its own magnitude. Therefore, when a major entries becomes significantly large, it also suppresses itself, leading to the slowing down of its growth. Note that this effect only slows down the growth but will not reverse the direction, since for major entries the suppression term is always smaller than the growth term, until $w_{i,i}$ becomes 1 where the growth and suppression terms are equal and the evolution stops. The terminal stage of the growth of major entries are characterized by Lemma D.5.

### 4.2.2 EXPLAINING MODEL BEHAVIOR

Recall that we have $f\left(\boldsymbol{U}(t); \widehat{\boldsymbol{x}}\right)_k = \sum_{p=1}^{s} w_{k,p}(t)\mu_p$. We now explain how the stage-wise evolution of Jacobian described in Sec. 4.2.1 determines the evolution of the model output.

**Generalization Order and Terminal Phase Slowing Down.** From the discussions in Sec. 4.2.1, by the end of the training, all the major entries converge to $1$ and all minor entries converge to $0$. The major entries grows in the order of corresponding $a_p$, which is determined by $\mu_p$ and $\sigma_p$, and slows down when approaching the terminal. This explains our observation that directions with larger $\mu_p$ and / or $\sigma_p$ is learned first, as well as the terminal phase slowing down.

**Swing-by Dynamics and Non-monotonic Loss Curve.** We argue that the Swing-by Dynamics and the non-monotonic loss curve is caused by the multi-stage major growth vs. minor growth / suppression process. Importantly, in certain configurations, minor entries growing towards larger absolute values (which is the incorrect solution) can lead to the decay of the OOD test loss, and cause an "illusion of generalizing" that the output trajectory is moving towards improving OOD generalization. However, this effect is later eliminated by the suppression of the corresponding minor entries, leading to a double (or multiple) descent-like loss curve and a reversal in the output trajectory.

More concretely, consider the first (initial) growth stage as an example. In this stage, for each $k \in [s]$, $f\left(\boldsymbol{U}(t); \widehat{\boldsymbol{x}}\right)_k$ is dominated by $w_{k,1}(t)\mu_k$, since $w_{k,1}$ grows fastest among all the entries in the $k$-th row. If $w_{k,1}$ happens to be initialized positive, then $f\left(\boldsymbol{U}(t); \widehat{\boldsymbol{x}}\right)_k$ grows towards $1$, which is the correct direction[4], and loss thus decays. Since in a symmetric initialization, each entry has equal chance of being initialized positive or negative, when $s$ is small, it is easy to have many minor entries initialized positive, whose growth contributes to the decaying of loss. *This causes an illusion that the model is going towards the right direction of OOD generalization.* After the minor entries of the first group are suppressed, their contribution to the decaying of the loss is canceled, which leads to the output trajectory turning back to the direction of memorizing a training cluster and a transient loss increase.

Fig. 5 presents the loss curve and the Jacobian entry evolution predicted by the theory with a specific initialization. Notice how, as claimed above, the first and second descending of loss accurately corresponds to the initial and second growth of the major entries, and the ascending of the loss corresponds to the suppression of the minor entries. When $s > 2$, there are multiple turns of growth and suppression stages and can possibly leads to a multiple-descent-like loss curve, which we confirm and illustrate in App. E.1.

**Remark on the Role of Out-of-Distribution.** If one of the training cluster is very close to the test point, i.e. there is an $p \in [s]$ where $\sigma_p \gtrsim \mu_p$, then the setting becomes highly in-distribution, and intuitively there shouldn't be a significant Swing-by since the training loss monotonically decays. We note that this intuition is captured by our theory through Assumption D.5, which requires that the signal strength of different directions to be distinct enough, and this essentially prevents a too large $\sigma_p$ (compared to $\mu_p$).

**Remark on Failure Modes.** We note that our theory also provides an explanation on instances when the model fails to achieve OOD generalization when one or more of our assumptions outlined in App. D.1 breakdown. A specific case is when a major entry $w_{k,k}$ is overly suppressed by a corresponding minor entry before it can begin to grow, causing the growth term $G_{k,k}$ becomes nearly zero. Consequently, the model output at $\widehat{\boldsymbol{x}}$ in that direction converges to $0$, instead of $\mu_k$ as expected. See App. E.3 for more discussions and illustrations.

**Remark on Existing Work.** There has been extensive research on the non-monotonic behavior of linear neural networks (in various settings). We note that existing studies either focus on one-layer networks (Pezeshki et al., 2022; Heckel & Yilmaz, 2020) or diagonally initialized networks (Lampinen & Ganguli, 2018; Saxe et al., 2019; Even et al., 2023; Pesme & Flammarion, 2023; Olmin & Lindsten, 2024), which essentially make the evolution of each direction decoupled. This decoupling simplifies the learning dynamics and can overlook critical aspects thereof (as we discussed in the preceding subsection). In contrast, our analysis, through a careful treatment of each entry of the Jacobian, does not need to make the diagonal initialization assumption, hence allowing

---

[4]Notice that this is true even when $k \neq 1$, i.e. $w_{k,1}$ is a minor entry.

us to capture and characterize the rich behaviors that arise from the interaction between different directions.

## 5 DIFFUSION MODEL RESULTS

Tying back to our original motivation of devising an abstraction of concept space first explored in text-to-image generative diffusion models, we now aim to verify if our findings and predictions made on SIM task can be reproduced in a more involved practical setup with diffusion models. To this end, we borrow the setup from Okawa et al. (2023); Park et al. (2024) and train conditional image diffusion models on two concepts—`size` and `color`.

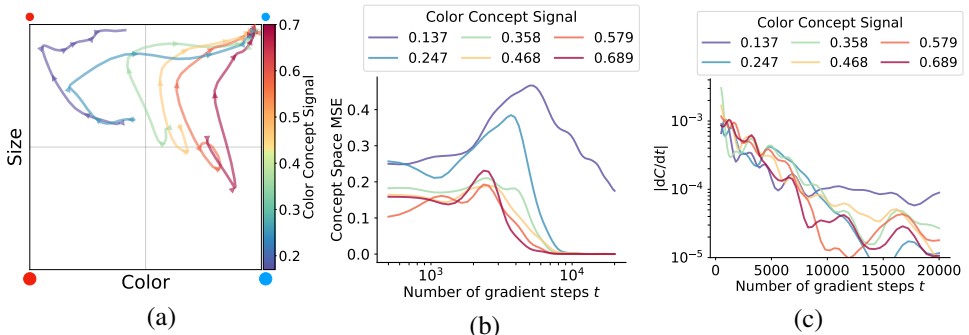

(a)  (b)  (c)

Figure 6: **Main observations reproduced on prompt-to-image diffusion models.** (a) Signal strength controls generalization speed and order; and a transient bend towards the concept with stronger signal. (b) A double descent-like curve for the concept space MSE caused by Swing-by Dynamics. (c) Concept learning gradually slows down. Our theory predicts speed of concept learning slows down at an exponential rate, which broadly matches the experimental results.

In Fig. 6, we repeat the experiments did in Section 4 of Park et al. (2024): we consider a synthetic setup where the model learns to generate the image of a circle of the indicated size and color given by a text input. The training set only contains samples from input pairs (`red`, `big`), (`blue`, `big`) and (`red`, `small`), and the test set contains samples from an OOD input pair (`blue`, `small`). A pretrained classier is applied to map the image generated by the diffusion model back to the concept space so that we can plot the model output curve in the concept space, as shown in Fig. 6 (a). In order to compare model behaviors under different signal strength, we tune the contrast between the color of red samples and blue samples in the training set to control the strength of the color signal (See Fig. 19 for an illustration). See App. G for experiment details.

Fig. 6 (a) shows that the level of concept signal largely alters the generalization dynamics. Specifically, we see that the order of compositional generalization is determined by the color concept signal, and can be reversed when we tuning the concept signal. Additionally, Fig. 6 (a) along with the corresponding loss curve plot Fig. 6 (b) also shows Swing-by Dynamics where the generalization dynamics show a bend towards the concept with stronger signal; the bend is transient and the generation eventually converges to the correct class (small blue circle), and the corresponding test loss curve shows a double descent-like trend. Fig. 6 (c) confirms that the speed of compositional generalization, quantified by the concept space traversal distance per step, decelerates at an exponential rate, as expected from our theoretical findings (Theorem 4.1).

## 6 CONCLUSION

In this paper, we propose SIM task as a further abstraction of the "concept space" previously explored by Okawa et al. (2023); Park et al. (2024). We conduct comprehensive investigation into the behaviors of a regression model trained on SIM, both empirically and theoretically, demonstrating that the learning dynamics on SIM effectively captures the phenomena observed on image generation task, establishing SIM as a basis for studying compositional generalization. Critically, our theoretical analysis uncovers the underlying causes of several phenomena that previously observed on compositional generalizations, as well as predicting new ones that characterizes the multi-stage and non-monotonic learning dynamics, which have been largely overlooked in earlier research. Our diffusion model experiments further verify the validity of our analysis. Additional discussions and potential future work directions can be found in App. E.

ACKNOWLEDGEMENTS

Part of this work was done while YY and WH were visiting the Simons Institute for the Theory of Computing.

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

# A  RELATED WORK

In this section, we provide some context for this paper by reviewing some existing work on compositional generalization and the study of deep linear networks.

**Compositional Generalization.**    Prior work on compositionality has often focused on benchmarking of pretrained models (Thrush et al., 2022; Andreas, 2019; Lewis et al., 2022; Lake & Baroni, 2018; Yun et al., 2022; Lepori et al., 2023; Johnson et al., 2017; Conwell & Ullman, 2022; Yuksekgonul et al., 2022; Schott et al., 2021; Gokhale et al., 2022; Valvoda et al., 2022) or proposition of protocols that allow generation of compositional samples (Du et al., 2021; 2023; Liu et al., 2022a; Xu et al., 2022; Yuksekgonul et al., 2022; Bugliarello & Elliott, 2021; Spilsbury & Ilin, 2022; Kumari et al., 2023; Du & Kaelbling, 2024). While perfect compositionality in natural settings is still lacking (Marcus et al., 2022; Leivada et al., 2022; Conwell & Ullman, 2022; 2023; Gokhale et al., 2022; Du et al., 2023; Liu et al., 2022a; Singh et al., 2021; Rassin et al., 2022; Feng et al., 2022; Hutchinson et al., 2022), several works have demonstrated via use of toy settings that this is unlikely to be an expressibility issue, as was hypothesized, e.g., by Fodor et al. (1975), since the model can in fact learn to perfectly compose in said toy settings. The ability to compose is in fact rather distinctly emergent (Okawa et al., 2023; Lubana et al., 2024) and the model learning it often correlates with distinctive patterns in the learning dynamics, as identified by Park et al. (2024). We note that there has in fact been some work on understanding compositional generalization abilities in neural networks (Wiedemer et al., 2024; Udandarao et al., 2024; Ramesh et al., 2023), but, unlike us, the focus of these papers is not on the model's learning dynamics.

**Learning Dynamics of Deep Linear Networks.**    Deep linear networks has been a commonly studied model for learning dynamics, and existing works mostly focus on the final solution found by the model, which primarily concerns the stationary point of the dynamics (Arora et al., 2018; Ji & Telgarsky, 2018; Du & Hu, 2019; Arora et al., 2019; Advani et al., 2020). There have also been works that try to characterize the full learning dynamics; however, they generally require the learning of each direction (neuron) to be decoupled (Saxe et al., 2013; Olmin & Lindsten, 2024; Lampinen & Ganguli, 2018; Even et al., 2023; Pesme & Flammarion, 2023; Pesme et al., 2021), which can be realized through a specific initialization choice. The decoupling assumption ignores the interaction between different neurons and highly simplify the dynamics, and as we mentioned in Sec. 4.1, make it unable to capture some important phenomena in practice. The symmetric 2-layer linear model is also a specific model that is frequently studied, especially in matrix sensing (Li et al., 2020; Stöger & Soltanolkotabi, 2021; Jin et al., 2023), and as we noted in Sec. 4.2, current theoretical results of this model focus on the implicit biases in the solutions learned, while our analysis, on the other hand, aims at characterizing the full learning dynamics and focus on its OOD behavior.

# B  MODEL COMPOSITIONALLY GENERALIZE IN TOPOLOGICALLY  CONSTRAINED ORDER

In this section, we introduce another phenomenon observed on SIM task learning that we do not put in the main paper: the order of compositional generalization happens in a topologically constrained order.

In this section, instead of the single test point $\widehat{x}$, we introduce a hierarchy of test points. Specifically, let $\mathcal{I} = \{0, 1\}^s$ be the index set of test points. For each $v \in \mathcal{I}$, we define a test point

$$\widehat{x}^{(v)} = \sum_{p=1}^{s} v_p \mu_p \mathbf{1}_p, \tag{B.1}$$

and call $\widehat{x}^{(v)}$ the test point with the index $v$. Intuitively, the index $v$ describes which training sets are combined into the current test point. If $\|v\| = 1$ then $\widehat{x}^{(v)}$ is the center of one of the training clusters.

We assign the component-wise ordering $\preceq$ to the index set $\mathcal{I}$, i.e., for $u, v \in \mathcal{I}$, we say $u \preceq v$ if and only if $\forall i \in [n], u_i \leq v_i$. It's easy to see that $\preceq$ is a partial-ordering.

Interestingly, in the SIM experiment, the order of the generalization in different test points strictly follow the component-wise order. This finding can be described formally in the following way: the

loss function is an order homomorphism between $\preceq$ on the index set, and $\leq$ on the real number. Let $\ell(z)$ be the loss function of the test point $z$, then we have the following empirical observation:

$$\forall \boldsymbol{u}, \boldsymbol{v} \in \mathcal{I}, \boldsymbol{u} \preceq \boldsymbol{v} \implies \ell\left(\tilde{\boldsymbol{x}}^{(u)}\right) \leq \ell\left(\tilde{\boldsymbol{x}}^{(v)}\right). \tag{B.2}$$

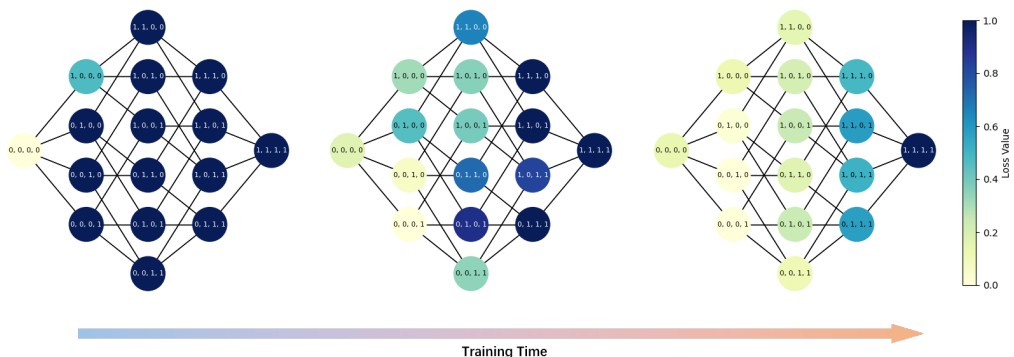

Figure 7: The loss at each test point in different timepoints during training for a 2-layer MLP with ReLU activation. Each graph represents a timepoint. Each node in the graph represents a test point, with index printed on it, and edges connecting nodes with Hamming distance 1. The color of the graph represents the loss of corresponding test point. Notice that we truncate the loss at 1 in order to unify the scale. From lest to right: epoch $= 1, 3, 5$.

In Fig. 7 we show the loss of each test point in several timepoints, with $\boldsymbol{\mu} = (1, 2, 3, 4)$, $\boldsymbol{\sigma} = \left\{\frac{1}{2}\right\}^4$. There is a clear trend that the test points that are on the right of the graph (larger in the component-wise order) will only be learned after all of its predecessors are all learned. We call this phenomenon the *topological constraint* since the constraint is based on the topology of the graph in Fig. 7.

## C  PROOFS AND CALCULATIONS

In the main text we have omitted some critical proofs and calculations due to space limitation. In this section we provide the complete derivations. Note that we postpone the proof of related theorems of Sec. 4.2 to App. D because of their length.

### C.1  THE LOSS FUNCTION WITH LINEAR MODEL AND INFINITE DATA LIMIT

In this subsection we derive the transformed loss function eq. (4.1), as well as the expression of the data matrix $\boldsymbol{A}$. For convenience we denote $\boldsymbol{W_\theta}$ by $\boldsymbol{W}$. We have

$$\mathcal{L}(\boldsymbol{\theta}) = \frac{1}{2ns} \sum_{p=1}^{s} \sum_{k=1}^{n} \left\| (\boldsymbol{W} - \boldsymbol{I}) \boldsymbol{x}_k^{(p)} \right\|^2 \tag{C.1}$$

$$= \frac{1}{2ns} \operatorname{Tr}\left[ \boldsymbol{x}_k^{(p)\top} (\boldsymbol{W} - \boldsymbol{I})^\top (\boldsymbol{W} - \boldsymbol{I}) \boldsymbol{x}_k^{(p)} \right] \tag{C.2}$$

$$= \frac{1}{2ns} \operatorname{Tr}\left[ (\boldsymbol{W} - \boldsymbol{I})^\top (\boldsymbol{W} - \boldsymbol{I}) \boldsymbol{x}_k^{(p)} \boldsymbol{x}_k^{(p)\top} \right] \tag{C.3}$$

$$= \frac{1}{2} \operatorname{Tr}\left[ (\boldsymbol{W} - \boldsymbol{I})^\top (\boldsymbol{W} - \boldsymbol{I}) \frac{1}{ns} \boldsymbol{x}_k^{(p)} \boldsymbol{x}_k^{(p)\top} \right] \tag{C.4}$$

$$= \frac{1}{2} \operatorname{Tr}\left[ \boldsymbol{A}^{1/2} (\boldsymbol{W} - \boldsymbol{I})^\top (\boldsymbol{W} - \boldsymbol{I}) \boldsymbol{A}^{1/2} \right] \tag{C.5}$$

$$= \frac{1}{2} \left\| (\boldsymbol{W} - \boldsymbol{I}) \boldsymbol{A}^{1/2} \right\|_{\mathcal{F}}^2. \tag{C.6}$$

Let $\mathcal{G}$ be the data generating process. It can be viewed as two components: first assign one of the $s$ clusters, and then draw a Gaussian vector from a Gaussian distribution in that cluster. Specifically,

let $\boldsymbol{x}$ be an arbitrary sample from the traning set, then the distribuition of $\boldsymbol{x}$ is equal to

$$\boldsymbol{x} \simeq \boldsymbol{\mu}^{(\eta)} + \operatorname{diag}(\boldsymbol{\sigma})\boldsymbol{\xi}, \tag{C.7}$$

where $\eta$ is a uniform random variable taking values in $[s]$ and $\boldsymbol{\xi} \sim \mathcal{N}(\boldsymbol{0}, \boldsymbol{I})$ is a random Gaussian vector that is independent from $\eta$. Here $\simeq$ represents having the same distribution.

When $n \to \infty$, the data matrix $\boldsymbol{A}$ converges to the true covariance, which is is

$$\boldsymbol{A} \to \mathbb{E}\left(\boldsymbol{x}\boldsymbol{x}^\top\right) \tag{C.8}$$

$$= \mathbb{E}\left[\left(\boldsymbol{\mu}^{(\eta)} + \operatorname{diag}(\boldsymbol{\sigma})\boldsymbol{\xi}\right)\left(\boldsymbol{\mu}^{(\eta)} + \operatorname{diag}(\boldsymbol{\sigma})\boldsymbol{\xi}\right)^\top\right] \tag{C.9}$$

$$= \mathbb{E}\left(\boldsymbol{\mu}^{(\eta)}\boldsymbol{\mu}^{(\eta)\top}\right) + \mathbb{E}\operatorname{diag}(\boldsymbol{\sigma})\boldsymbol{\xi}\boldsymbol{\xi}^\top\operatorname{diag}(\boldsymbol{\sigma}) \tag{C.10}$$

$$= \frac{1}{s}\sum_{p=1}^{s}\boldsymbol{\mu}^{(p)}\boldsymbol{\mu}^{(p)\top} + \operatorname{diag}(\boldsymbol{\sigma})^2 \tag{C.11}$$

$$= \frac{1}{s}\sum_{p=1}^{s}\mu_p^2\mathbf{1}_p\mathbf{1}_p^\top + \operatorname{diag}(\boldsymbol{\sigma})^2 \tag{C.12}$$

$$= \frac{1}{s}\operatorname{diag}(\boldsymbol{\mu})^2 + \operatorname{diag}(\boldsymbol{\sigma})^2. \tag{C.13}$$

## C.2 PROOF OF THEOREM 4.1

In this subsection for the notation-wise convenience we denote $\boldsymbol{W} = \boldsymbol{\theta}$. Since the model is one-layer, the loss function eq. (4.1) becomes

$$\mathcal{L}(\boldsymbol{W}) = \frac{1}{2}\left\|(\boldsymbol{W} - \boldsymbol{I})\boldsymbol{A}^{1/2}\right\|_{\mathcal{F}}^2, \tag{C.14}$$

and the gradient is

$$\nabla\mathcal{L}(\boldsymbol{W}) = (\boldsymbol{W} - \boldsymbol{I})\boldsymbol{A} = \boldsymbol{W}\boldsymbol{A} - \boldsymbol{A}. \tag{C.15}$$

We denote the $k$-th row of $\boldsymbol{W}$ and $\boldsymbol{A}$ by $\boldsymbol{w}_k$ and $\boldsymbol{A}_k$ respectively. Then we have

$$\dot{\boldsymbol{w}}_k = -\boldsymbol{A}\boldsymbol{w}_k + \boldsymbol{a}_k. \tag{C.16}$$

The solution of this differential equation is

$$\boldsymbol{w}_k(t) = \exp\left(-\boldsymbol{A}t\right)\left[w_k(0) - \boldsymbol{A}^{-1}\boldsymbol{a}_k\right] + \boldsymbol{A}^{-1}\boldsymbol{a}_k, \tag{C.17}$$

where we use the convention $0 \times \left(0^{-1}\right) = 0$ to avoid the non-invertible case of $\boldsymbol{A}$.

Thus for any $\boldsymbol{z} \in \mathbb{R}^d$ we have

$$f(\boldsymbol{W}(t); \boldsymbol{z})_k = \langle \boldsymbol{w}_k(t), \boldsymbol{z}\rangle \tag{C.18}$$

$$= \left\langle\left(\boldsymbol{I} - e^{-\boldsymbol{A}t}\right)\boldsymbol{A}^{-1}\boldsymbol{a}_k, \boldsymbol{z}\right\rangle + \left\langle e^{-\boldsymbol{A}t}w_k(0), \boldsymbol{z}\right\rangle \tag{C.19}$$

$$= \sum_{p=1}^{n}\frac{1 - e^{-a_p t}}{a_p}\mathbb{1}_{\{k=p\}}a_p z_p + \sum_{i=1}^{n}e^{-a_i t}w_{k,i}(0)z_i \tag{C.20}$$

$$= \mathbb{1}_{\{k \leq s\}}\left(1 - e^{-a_k t}\right)z_k + \sum_{i=1}^{n}e^{-a_i t}w_{k,i}(0)z_i, \tag{C.21}$$

and this proves the claim.

## D THEORETICAL ANALYSIS OF THE TWO LAYER MODEL

In this section we provide a detailed analysis of the symmetric two-layer linear model described in Sec. 4.2.

In this section we assume a finite step size, i.e., $\boldsymbol{W} : \mathbb{N} \to \mathbb{R}^{d \times d}$ is initialized by $\boldsymbol{W}(0)$ and updated by

$$\frac{\boldsymbol{W}(t+1) - \boldsymbol{W}(t)}{\eta} = -\boldsymbol{U}(t)\nabla\mathcal{L}(\boldsymbol{U}(t))^\top - \nabla\mathcal{L}(\boldsymbol{U}(t))\boldsymbol{U}(t)^\top \tag{D.1}$$

$$= \boldsymbol{W}(t)\boldsymbol{A} + \boldsymbol{A}\boldsymbol{W}(t) - \frac{1}{2}\left[\boldsymbol{A}\boldsymbol{W}(t)^2 + \boldsymbol{W}(t)^2\boldsymbol{A} + 2\boldsymbol{W}(t)\boldsymbol{A}\boldsymbol{W}(t)\right]. \tag{D.2}$$

The update of each entry $w_{i,j}(t)$ can be decomposed into three terms, as we described in the main text:

$$\frac{w_{i,j}(t+1) - w_{i,j}(t)}{\eta} = w_{i,j}(t)(a_i + a_j) - \frac{1}{2}\sum_{k=1}^{d} w_{k,i}w_{k,j}(a_i + a_j + 2a_k) \tag{D.3}$$

$$= \underbrace{w_{i,j}(t)(a_i + a_j)}_{G_{i,j}(t)} \tag{D.4}$$

$$\underbrace{-\frac{1}{2}w_{i,j}\left[w_{i,i}(3a_i + a_j) + \mathbb{1}_{\{i\neq j\}}w_{j,j}(3a_j + a_i)\right]}_{S_{i,j}(t)} \tag{D.5}$$

$$\underbrace{-\frac{1}{2}\sum_{\substack{k\neq i \\ k\neq j}} w_{k,i}(t)w_{k,j}(t)(a_i + a_j + 2a_k)}_{N_{i,j}(t)}. \tag{D.6}$$

### D.1 ASSUMPTIONS

We need make several assumptions to prove the results. Below we make several assumptions that all commonly hold in the practice. The first assumption to make is that both the value of $a_k$ and the initialization of $\boldsymbol{W}$ is bounded.

**Assumption D.1** (Bounded Initialization and Signal Strength). *There exists $\alpha > 0, \gamma > 1, \beta > 1$ such that*

$$\forall k, \alpha \leq a_k \leq \gamma\alpha, \tag{D.7}$$

$$\forall i, j, \omega \leq |w_{i,j}(0)| \leq \beta\omega. \tag{D.8}$$

The second assumption is that the step size is small enough.

**Assumption D.2** (Small Step Size). *There exists a constant $K \geq 20$, such that $\eta \leq \frac{1}{9K\gamma\alpha}$.*

Next, we define a concept called initial phase. The definition of initial phase is related to a constant $P > 0$.

**Definition D.1.** *Assume there is a constant $P > 0$. For an entry $(i, j)$ and time $t$, if $|w_{i,j}(t)| \leq P\beta\omega$, we say this entry is in **initial phase**.*

As the name suggests, in the initial phase, the entries shouldn't be too far away from the initialization, and we achieve this by an upper bound of $P$.

**Assumption D.3** (Small Initial Phase). $P\omega\beta \leq 0.4$.

The next assumption to make is that the initialization value (i.e. $\omega$) should not be too large.

**Assumption D.4** (Small Initialization).

$$\omega \leq \min\left\{\frac{\min\{\kappa - 1, 1 - \kappa^{-1/2}\}}{PK\gamma d\beta^2}, \frac{1}{\sqrt{2\beta}}\right\} \tag{D.9}$$

*and $\kappa > 1.1$, and $\kappa \leq 1 + \frac{1}{2}KC^{-1}$, $P \geq 2$.*

Finally, we also assume that the signal strength difference is significant enough.

**Assumption D.5** (Significant Signal Strength Difference). *For any $i > j$, we have*

$$\frac{a_i + a_j}{2a_i} \leq \frac{\log P}{10\kappa^2 \log\frac{1}{P\beta\omega} + \log P\beta}. \tag{D.10}$$

*and there exists a constant $C > 1$ such that $a_i - 3a_j \geq C^{-1}\alpha$.*

## D.2 THE CHARACTERIZATION OF THE EVOLUTION OF THE JACOBIAN

In this subsection, we provide a series of lemmas that characterize each stage the evolution of the Jacobian matrix $W$.

The whole proof is based on induction, and in order to avoid a too complicated induction, we make the following assertion, which obviously holds at initialization.

**Assertion D.1.** *For all $t \in \mathbb{N}$, if $i \neq j$, then the entry $(i, j)$ stays in the initial phase for all time.*

We will use Assertion D.1 as an assumption throughout the proves and prove it at the end. This is essentially another way of writing inductions.

We have the following corollary that directly followed by Assertion D.1.

**Corollary D.1.** *For all $t \in \mathbb{N}$ and all $i, j$, $|N_{i,j}(t)| \leq 2P\gamma\alpha d\beta^2\omega^2$.*

Now, we are ready to present and prove the major lemmas. The first lemma is to post a (rather loose) upper bound of the value of the entries.

**Lemma D.1** (Upper Bounded Growth). *Consider entry $(i, j)$. We have for all $t \in \mathbb{N}$, at timepoint $t$ the absolute value of the $(i, j)$-th entry satisfies*

$$|w_{i,j}(t)| \leq |w_{i,j}(0)| \exp\left[\eta t(a_i + a_j)\kappa\right]. \tag{D.11}$$

*Proof.* Since of the $N_{i,j}$ term we only use its absolute value, the positive case and negative case are symmetric. WLOG we only consider the case where $w_{i,j}(0) > 0$ here.

The claim is obviously satisfied at initialization. We use it as the inductive hypothesis. Suppose at timepoint $t \leq T - 1$ the claim is satisfied, we consider the time step $t + 1$.

Since Assertion D.1 guaranteed that every non-diagonal entry is in the initial phase, and the $S_{i,j}$ term has different symbol with $w_{i,j}(0)$, we have

$$|S_{i,j}(t) + N_{i,j}(t)| \leq 2P\gamma\alpha d\beta^2\omega^2. \tag{D.12}$$

We have

$$w_{i,j}(t+1) - w_{i,j}(t) \leq \eta w_{i,j}(t)(a_i + a_j) + 2P\eta\gamma\alpha d\beta^2\omega^2 \tag{D.13}$$

$$\leq \eta(a_i + a_j)w_{i,j}(0)\exp\left[\eta t(a_i + a_j)\kappa\right] + 2P\eta\gamma\alpha d\beta^2\omega^2 \tag{D.14}$$

$$= w_{i,j}(0)\exp\left[\eta t(a_i + a_j)\kappa\right]\left[\eta(a_i + a_j) + \frac{2P\eta\gamma\alpha d\beta^2\omega^2}{w_{i,j}(0)\exp\left[\eta t(a_i + a_j)\kappa\right]}\right] \tag{D.15}$$

From Assumption D.4, we have

$$\eta(a_i + a_j) + \frac{2P\eta\gamma\alpha d\beta^2\omega^2}{w_{i,j}(0)\exp\left[\eta t(a_i + a_j)\kappa\right]} \leq \eta(a_i + a_j) + 2P\gamma\alpha d\beta^2\omega \tag{D.16}$$

$$\leq \eta(a_i + a_j) + 2(\kappa - 1)\eta\alpha \tag{D.17}$$

$$\leq \kappa\eta(a_i + a_j) \tag{D.18}$$

$$\leq \exp(\kappa\eta[a_i + a_j]) - 1, \tag{D.19}$$

thus we have

$$w_{i,j}(t+1) \leq w_{i,j}(t) + \left[\exp(\kappa\eta[a_i + a_j]) - 1\right]w_{i,j}(t) \tag{D.20}$$

$$\leq w_{i,j}(0)\exp\left[\eta(t+1)(a_i + a_j)\kappa\right]. \tag{D.21}$$

$\square$

Next, we prove that Lemma D.1 is tight in the initial stage of the training, up to a constant $\kappa$ in the exponential term.

**Lemma D.2** (Lower Bounded Initial Growth). *Let $T_1 = \frac{\log P}{2\eta\gamma\alpha\kappa}$. We have for all $t \in [T_1]$, at timepoint $t$ every entry $(i, j)$ is in the initial phase, and the absolute value of the $(i, j)$-th entry satisfies*

$$|w_{i,j}(t)| \geq |w_{i,j}(0)| \exp\left[\eta t(a_i + a_j)\kappa^{-1}\right] \qquad (D.22)$$

*and $w_{i,j}(t)w_{i,j}(0) > 0$.*

*Proof.* Similar to the proof of Lemma D.1, we may just assume $w_{i,j}(0) > 0$.

Moreover, we also use the claim as an inductive hypothesis and prove it by induction. Since here the inductive hypothesis states that every entry is in the initial phase, we have

$$|S_{i,j}(t) + N_{i,j}(t)| \leq 4\gamma\alpha d\beta^2\omega^2. \qquad (D.23)$$

We have

$$w_{i,j}(t + 1) - w_{i,j}(t) \geq \eta(a_i + a_j)w_{i,j}(0)\exp\left[\eta t(a_i + a_j)\kappa^{-1}\right] - 2P\eta\gamma\alpha d\beta^2\omega^2 \qquad (D.24)$$

$$= w_{i,j}(0)\exp\left[\eta t(a_i + a_j)\kappa^{-1}\right]\left[\eta(a_i + a_j) - \frac{2P\eta\gamma\alpha d\beta^2\omega^2}{w_{i,j}(0)\exp\left[\eta t(a_i + a_j)\kappa^{-1}\right]}\right] \qquad (D.25)$$

From Assumption D.4, we have

$$\frac{2P\eta\gamma\alpha d\beta^2\omega^2}{w_{i,j}(0)\exp\left[\eta t(a_i + a_j)\kappa^{-1}\right]} \leq 2P\eta\gamma\alpha d\beta^2\omega \qquad (D.26)$$

$$\leq \left(1 - \kappa^{-1/2}\right)\eta(a_i + a_j). \qquad (D.27)$$

Moreover, notice that when $\kappa > 1.1$, for any $x < 0.1$, we have $\kappa^{-1/2}x + 1 \geq e^{\kappa^{-1}x}$. Since Assumption D.2 ensured that $\eta \leq \frac{1}{10(a_i + a_j)}$, we have

$$w_{i,j}(t + 1) \geq w_{i,j}(t) + w_{i,j}(t)\left[\kappa^{-1/2}\eta(a_i + a_j)\right] \qquad (D.28)$$

$$\geq w_{i,j}(t)\exp\left(\eta(a_i + a_j)\kappa^{-1}\right) \qquad (D.29)$$

$$\geq w_{i,j}(0)\exp\left[\eta(t + 1)(a_i + a_j)\kappa^{-1}\right]. \qquad (D.30)$$

Finally, from Lemma D.1, we have when

$$w_{i,j}(t) \leq |w_{i,j}(0)|\exp\left(\eta t(a_i + a_j)\kappa\right) \qquad (D.31)$$

$$\leq \beta\omega\exp\left(2\eta T_1\gamma\alpha\kappa\right) \qquad (D.32)$$

$$= P\beta\omega, \qquad (D.33)$$

which confirms that every entry $(i, j)$ stays in the initial phase before time $T_1$.

$\square$

Notice that the time bound $T_1$ in Lemma D.2 is a uniform one which applies to all entries. For the major entries, we might want to consider a finer bound of the time that it leaves the initial phase. This can be proved by essentially repeating the same proof idea of Lemma D.2.

**Lemma D.3** (Lower Bounded Initial Growth for Diagonal Entries). *Consider an diagonal entry $(i, i)$. Let $T_1^{(i)} = \frac{\log\frac{P\beta\omega}{w_{i,i}(0)}}{2\eta a_i\kappa}$. We have for all $t \in \left[T_1^{(i)}\right]$, at timepoint $t$ the entry $(i, i)$ is in the initial phase, and the absolute value of the $(i, i)$-th entry satisfies*

$$w_{i,i}(t) \geq w_{i,i}(0)\exp\left(2\eta t a_i\kappa^{-1}\right). \qquad (D.34)$$

We omit the proof of Lemma D.3 since it is almost identical to the proof of Lemma D.2, only with replacing $\gamma\alpha$ by $a_i$ and $\beta\omega$ by $w_{i,i}(0)$.

Next, we characterize the behavior of one diagonal entry after it leaves the initial phase.

**Lemma D.4** (Lower Bounded After-Initial Growth for Diagonal Entries). *Consider a diagonal entry* $(i, i)$. *If at time* $t_0$ *we have* $|w_{i,i}(t_0)| \geq P\beta\omega$, *and for a* $\lambda \in (P\beta\omega, 1 - K^{-1})$, *before time* $T^{(\lambda)}$ *we have* $w_{i,i}(t + t_0) < \lambda$ *for all* $t \in [T^{(\lambda)}]$, *then we have*

$$w_{i,i}(t + t_0) \geq w_{i,i}(t_0) \exp\left[2\eta t a_i (1 - \lambda)\kappa^{-1}\right]. \tag{D.35}$$

*Moreover,* $w_{i,i}(0), w_{i,i}(t_0), w_{i,i}(t_0 + t) \geq 0$.

*Proof.* Notice that since $\boldsymbol{W} = \boldsymbol{U}\boldsymbol{U}^\top$ is a PSD matrix, its diagonal entries are always non-negative, this ensures that $w_{i,i}(0), w_{i,i}(t_0), w_{i,i}(t_0 + t) \geq 0$.

For the time after $t_0$ and before $t_0 + T^{(\lambda)}$, we use an induction to prove the claim, with the claim itself as the inductive hypothesis. It clearly holds when $t = 1$.

Notice that when $w_{i,j}(t') < \lambda$, we have

$$G_{i,j}(t') - S_{i,j}(t') = 2a_i w_{i,i}(t') \left[1 - w_{i,i}(t')\right] \geq 2a_i w_{i,i}(t')(1 - \lambda). \tag{D.36}$$

Thus we have

$$w_{i,i}(t_0 + t + 1) - w_{i,i}(t_0 + t) \tag{D.37}$$

$$\geq 2\eta a_i (1 - \lambda) w_{i,i}(t_0) \exp\left[\eta t (a_i + a_j)(1 - \lambda)\kappa^{-1}\right] - 2P\eta\gamma\alpha d\beta^2\omega^2 \tag{D.38}$$

$$= w_{i,i}(t_0) \exp\left[2\eta t a_i (1 - \lambda)\kappa^{-1}\right] \left[2\eta a_i (1 - \lambda) - \frac{2P\eta\gamma\alpha d\beta^2\omega^2}{w_{i,i}(t_0) \exp\left[2\eta t a_i (1 - \lambda)\kappa^{-1}\right]}\right] \tag{D.39}$$

Since $\lambda < 1 - K^{-1}$, and $w_{i,i}(t_0) \geq 2\beta\omega \geq \omega$, from Assumption D.4, we have

$$\frac{2P\eta\gamma\alpha d\beta^2\omega^2}{w_{i,i}(t_0) \exp\left[2\eta t a_i (1 - \lambda)\kappa^{-1}\right]} \leq 2P\eta\gamma\alpha d\beta^2\omega \tag{D.40}$$

$$\leq 2K^{-1}\left(1 - \kappa^{-1/2}\right)\eta\alpha \tag{D.41}$$

$$\leq 2\left(1 - \kappa^{-1/2}\right)\eta a_i (1 - \lambda). \tag{D.42}$$

Moreover, since Assumption D.2 ensured that $\eta \leq \frac{1}{2Ka_i(1-\lambda)} \leq \frac{1}{20a_i(1-\lambda)}$, using the fact that if $\kappa > 1.1$ then $\kappa^{-1/2}x + 1 \geq e^{\kappa^{-1}x}$ for any $x < 0.1$, we can get

$$w_{i,i}(t + 1) \geq w_{i,i}(t) + w_{i,i}(t) \left[\kappa^{-1/2} 2\eta a_i (1 - \lambda)\right] \tag{D.43}$$

$$\geq w_{i,i}(t) \exp\left(2\eta a_i \kappa^{-1}(1 - \lambda)\right) \tag{D.44}$$

$$\geq w_{i,i}(t_0) \exp\left[2\eta(t + 1)\kappa^{-1}(1 - \lambda)\right]. \tag{D.45}$$

$\square$

Next, we provide an uniform upper bound (over time) of the diagonal entries. Remember that we mentioned in the gradient flow case, the diagonal term stops evolving when it reaches 1. In the discrete case, since the step size is not infinitesimal, Lemma D.5 shows that it can actually exceed 1 a little bit but not too much since the step size is small.

**Lemma D.5** (Upper Bounded Diagonal Entry). *For any diagonal entry* $(i, i)$ *and any time* $t$, $0 \leq w_{i,i}(t) \leq 1 + 2K^{-1}$.

*Proof.* First notice that since $\boldsymbol{W}(t)$ is PSD, its diagonal entry $w_{i,i}(t)$ should always be non-negative, thus $w_{i,i}(t) \geq 0$ is always satisfied. In the following we prove $w_{i,i}(t) \leq 1 + 2K^{-1}$.

We use induction to prove this claim. The inductive hypothesis is the claim it self. It is obviously satisfied at initialization. In the following we assume the claim is satisfied at timepoint $t$ and prove it for timepoint $t + 1$. Notice that since $K \leq 10$, we have $1 + K^{-1} \leq 1 + 2K^{-1} \leq 2$.

Notice that by Assertion D.1 and Assumption D.4,

$$|N_{i,i}(t)| \leq 2P\gamma\alpha d\beta^2\omega^2 \leq \frac{(\kappa - 1)^2}{K^2\gamma d\beta^2}\alpha \leq K^{-1}a_i. \tag{D.46}$$

If $w_{i,i}(t) \geq 1 + K^{-1}$, we have

$$G_{i,i}(t) - S_{i,i}(t) = 2a_i w_{i,i}(1 - w_{i,i}) \leq -2a_i K^{-1}. \tag{D.47}$$

Therefore,

$$w_{i,i}(t+1) = w_{i,i}(t) + \eta \left[ G_{i,i}(t) - S_{i,i}(t) - N_{i,i}(t) \right] \tag{D.48}$$
$$\leq w_{i,i}(t) - a_i K^{-1} \eta \tag{D.49}$$
$$\leq w_{i,i}(t) \tag{D.50}$$
$$\leq 1 + 2K^{-1}. \tag{D.51}$$

Moreover, since $w_{i,i}(t) \leq 1 + 2K^{-1} \leq 2$, we have

$$|G_{i,i}(t)| + |S_{i,i}(t)| + |N_{i,i}(t)| \leq 4a_i + 4a_i + K^{-1}a_i \leq 9\gamma\alpha \leq \frac{1}{K\eta}. \tag{D.52}$$

When $w_{i,i}(t) \leq 1 + K^{-1}$, using eq. (D.52), we have

$$w_{i,i}(t+1) \leq w_{i,i}(t) + \eta \left( |G_{i,i}(t)| + |S_{i,i}(t)| + |N_{i,i}(t)| \right) \leq 1 + 2K^{-1}. \tag{D.53}$$

The above results together shows that $w_{i,i}(t+1) \leq 1 + 2K^{-1}$.

$\square$

**Corollary D.2** (Upper Bounded Diagonal Update)**.** *For any diagonal entry $(i, i)$ and any time $t$, $|w_{i,i}(t+1) - w_{i,i}(t)| \leq K^{-1}$.*

Corollary D.2 is a direct consequence of Lemma D.5 (and we actually proved Corollary D.2 in the proof of Lemma D.5).

The next lemma lower bounds the final value of diagonal entries. Together with Lemma D.5 we show that in the terminal stage of training the diagonal entries oscillate around 1 by the amplitude not exceeding $2K^{-1}$.

**Lemma D.6.** *Consider a diagonal entry $(i, i)$. If at time $t_0$ we have $w_{i,i}(t_0) \geq 1 - 2K^{-1}$, then for all $t' \geq t_0$ we have $w_{i,i}(t') \geq 1 - 2K^{-1}$.*

*Proof.* We use an induction here. The inductive hypothesis is the claim itself. This obviously holds when $t' = t_0$. We assume $w_{i,i}(t') \geq 1 - 2K^{-1}$ at timepoint $t'$ and prove the claim for $t' + 1$.

If $w_{i,i}(t') < 1 - K^{-1}$, then from Lemma D.4 we know

$$w_{i,i}(t'+1) \geq w_{i,i}(t') \geq 1 - 2K^{-1}. \tag{D.54}$$

If $w_{i,i}(t') > 1 - K^{-1}$, then from Corollary D.2 we have

$$w_{i,i}(t'+1) \geq w_{i,i}(t') - K^{-1} \geq 1 - 2K^{-1}. \tag{D.55}$$

$\square$

Now, we are ready to prove Assertion D.1 by considering the suppression. We first prove a lemma that upper bounds the absolute value of the minor entries after its corresponding major entry becomes significant.

**Lemma D.7** (Suppression)**.** *Consider an off-diagonal entry $(i, j)$ where $i > j$. If there exists a time $t_0$ such that $w_{i,i}(t_0) > 0.8$, then for any $t' \geq t_0$ we have*

$$|w_{i,j}(t')| \leq \max \{ |w_{i,j}(t_0)|, \omega \}. \tag{D.56}$$

*Proof.* Since $K > 10$, from Lemma D.6 and Lemma D.4 we know $w_{i,i}(t') > 0.8$ for all $t' \geq t_0$.

In this proof, we use an induction with the inductive hypothesis being the claim itself, i.e., we assume the claim is true at timepoint $t'$ and prove it for $t' + 1$. The claim obviously holds for $t' = t_0$.

Since in this proof we only use the absolute value of $N_{i,j}$, WLOG we may assume that $w_{i,j}(t') > 0$.

If $w_{i,j}(t') < \omega$ then we have proved the claim. In the following we may assume $w_{i,j}(t') \geq \omega$.

We have

$$G_{i,j}(t') - S_{i,j}(t') \leq w_{i,j}(t')(a_i + a_j) - \frac{1}{2}w_{i,j}(t')w_{i,i}(3a_i + a_j) \tag{D.57}$$

$$\leq w_{i,j}(t')(a_i + a_j) - w_{i,j}(t')\left[0.4(3a_i + a_j)\right] \tag{D.58}$$

$$= -\frac{1}{5}w_{i,j}(t')a_i + \frac{3}{5}w_{i,j}(t')a_j \tag{D.59}$$

$$\overset{(i)}{\leq} -C^{-1}\omega\alpha, \tag{D.60}$$

where in (i) we use Assumption D.5.

Thus we have

$$G_{i,j}(t') - S_{i,j}(t') - N_{i,j}(t') \leq G_{i,j}(t') - S_{i,j}(t') + |N_{i,j}(t')| \tag{D.61}$$

$$\leq -C^{-1}\omega\alpha + 2P\gamma\alpha d\beta^2\omega^2 \tag{D.62}$$

$$\overset{(i)}{<} 0, \tag{D.63}$$

where (i) is from Assumption D.4 and Assumption D.5. This confirms that $w_{i,j}(t'+1) < w_{i,j}(t') \leq \max\{|w_{i,j}(t_0), \omega\}$.

Next, we prove $w_{i,j}(t'+1) \geq -\max\{|w_{i,j}(t_0)|, \omega\}$. Notice that Lemma D.5 stated that $|w_{i,i}| \leq 2$. Notice that we also have $w_{i,j}(t') \leq K^{-1}$, thus from Assumption D.2,

$$|G_{i,j}(t')| + |S_{i,j}(t')| + |N_{i,j}(t')| \leq 10\gamma\alpha|w_{i,j}(t')| + 2P\gamma\alpha d\beta^2\omega^2 \tag{D.64}$$

$$\leq \frac{10|w_{i,j}(t')| + 2Pd\beta^2\omega^2}{9K\eta} \tag{D.65}$$

$$\leq \frac{10|w_{i,j}(t')| + 2\omega}{9K\eta} \tag{D.66}$$

$$\leq \frac{|w_{i,j}(t')| + \omega}{2\eta}. \tag{D.67}$$

We have

$$w_{i,j}(t'+1) \geq w_{i,j}(t') - \eta(|G_{i,j}(t')| + |S_{i,j}(t')| + |N_{i,j}(t)'|) \tag{D.68}$$

$$\geq -\eta(|G_{i,j}(t')| + |S_{i,j}(t')| + |N_{i,j}(t')|) \tag{D.69}$$

$$\geq -\frac{1}{2}(|w_{i,j}(t')| + \omega) \tag{D.70}$$

$$\geq -\max\{|w_{i,j}(t')|, \omega\}. \tag{D.71}$$

$$\square$$

With all the lemmas proved above, we are now ready to prove Assertion D.1.

**Lemma D.8** (Assertion D.1). *For all $t \in \mathbb{N}$, if $i \neq j$, then the entry $(i, j)$ stays in the initial phase for all time.*

*Proof.* Notice that since $W$ is symmetric, we only need to prove the claim for $i > j$. Moreover, From Lemma D.7, we only need to prove that there exists a timepoint $t^*$, such that $w_{i,i}(t^*) \geq 0.8$, and $|w_{i,j}(t^*)| \leq P\beta\omega$.

Let $t_0 = \frac{\log\frac{P\beta\omega}{w_{i,i}(0)}}{2\eta a_i\kappa}$, by Lemma D.3, we have $w_{i,i}(t_0) \geq P\beta\omega$. By Lemma D.3 and Lemma D.4, we have for any $t \geq t_0$ such that $w_{i,i}(t) \leq \lambda$, where $\lambda = 0.85$,

$$w_{i,i}(t) \geq w_{i,i}(t_0)\exp\left[0.3\eta(t - t_0)a_i\kappa^{-1}\right] \tag{D.72}$$

$$\geq P\beta\omega\exp\left[0.3\eta(t - t_0)a_i\kappa^{-1}\right] \tag{D.73}$$

Let $t'$ be the first time that $w_{i,i}(t')$ arrives above 0.8. Let $t^* = \min\left\{\frac{\kappa \log \frac{0.8}{P\beta\omega}}{0.3\eta a_i} + t_0, t'\right\} \geq t_0$. If $t^* = t'$, we have $w_{i,i}(t^*) \geq 0.8$. If $t^* = \frac{\kappa \log \frac{0.8}{P\beta\omega}}{0.3\eta a_i} + t_0$, we have

$$w_{i,i}(t^*) \geq w_{i,i}(0) \exp\left(0.3\eta t^* a_i \kappa^{-1}\right) \tag{D.74}$$

$$\geq P\beta\omega \exp\left(\log\frac{0.8}{P\beta\omega}\right) \tag{D.75}$$

$$= 0.8. \tag{D.76}$$

Moreover, from Lemma D.1 and Assumption D.5, we have

$$|w_{i,j}(t^*)| \leq |w_{i,j}(0)| \exp\left[\eta t^* \kappa(a_i + a_j)\right] \tag{D.77}$$

$$\leq \beta\omega \exp\left[\left(\frac{\kappa^2 \log \frac{0.8}{P\beta\omega}}{0.15} + \log\frac{P\beta\omega}{w_{i,i}(0)}\right) \times \frac{a_i + a_j}{2a_i}\right] \tag{D.78}$$

$$\leq \beta\omega \exp\left[\left(10\kappa^2 \log\frac{1}{P\beta\omega} + \log P\beta\right) \times \frac{a_i + a_j}{2a_i}\right] \tag{D.79}$$

$$\leq \beta\omega \exp\left[\log(P)\right] \tag{D.80}$$

$$= P\omega\beta. \tag{D.81}$$

The claim is thus proved by combining the above bounds on $|w_{i,j}(t^*)|$ and $w_{i,i}(t^*)$ with Lemma D.7. □

# E   ADDITIONAL DISCUSSIONS

In this section, we further discuss the findings and theoretical predictions presented in this paper.

## E.1   MULTIPLE DESCENTS

In Fig. 5, we verified our theoretical predictions of the Swing-by Dynamics through an experiment of an $s = 2$ example. However, in our theory, there can be multiple growth / suppression stages when $s > 2$, which should give us a multiple descent-like curve. We note here that based on the conditions given in App. D.1, it is indeed possible to see multiple descent but only with a subtle choice of the signal strengths ($\mu$) and under specific initialization conditions.

In Fig. 8 and 9, we illustrate two settings where the loss curves exhibit epoch-wise triple and quadruple descent. In both settings we use symmetric 2-layer linear model, same as the model used in Sec. 4.2. Note that we tuned initialization random seed to generate these results. Moreover, since the time scale of each descent vary, we use a log scale for the number of epochs to make the results more apparent.

It is worth noting that in Fig. 8 and 9, each major entry starts to grow only after the corresponding minor entry is suppressed (for example in Fig. 8, $w_{2,2}$ starts to grow after $w_{2,3}$ is suppressed, $w_{1,2}$ starts to decay after $w_{2,2}$ is close to 1, and $w_{1,1}$ starts to grow after $w_{1,2}$ is suppressed), and each ascending / descending stage of loss curve aligns well with a stage of growth / suppression of the minor and major entries. These correspondence match exactly with our theoretical prediction and shows the correctness and preciseness of our theory.

## E.2   THE BREAKDOWN OF THE INITIALIZATION ASSUMPTION

In Sec. 3.3, we mentioned that Swing-by Dynamics seems to be more significant when the dimensionality of the dataset is low, and in Sec. 4.2.2, we attributed the reason of it to the fact that when the dimensionality of the dataset is small, it's easier to have more minor entries initialized positive, which lead to an illusion of learning in the minor entry growth stage, whose later suppression leads to the non-monotonic output trajectory behavior of Swing-by Dynamics.

In this section, we note that, another reason for the Swing-by Dynamics to be less significant is that Assumption D.4 breaks down when the dimension is high, if we use standard Gaussian initialization

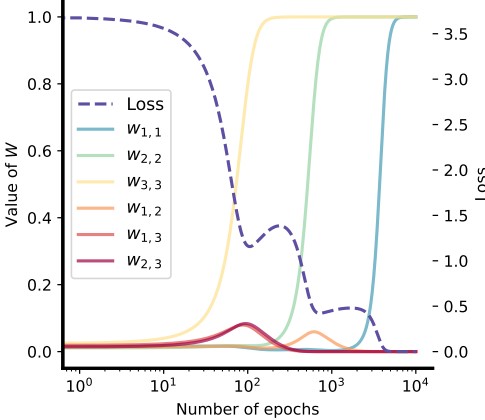 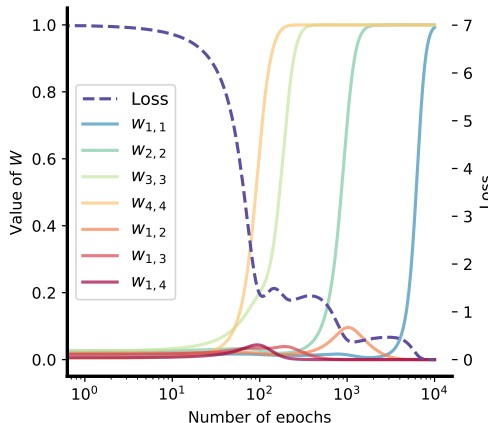

Figure 8: **An illustration of epoch-wise triple descent of the symmetric 2-layer linear model.** The dataset has dimensionality and number of informative directions $d = s = 3$, signal strength values $\boldsymbol{\mu} = (1.0, 1.5, 2.2)$, and noise values $\boldsymbol{\sigma} = (0.05, 0.05, 0.05)$.

Figure 9: **An illustration of epoch-wise quadruple descent with the symmetric 2-layer linear model.** The dataset has dimensionality and number of informative directions $d = s = 4$, signal strength values $\boldsymbol{\mu} = (1.0, 1.5, 2.2, 2.7)$, and noise values $\boldsymbol{\sigma} = (0.05, 0.05, 0.05, 0.5)$.

to initialize the model weights. Specifically, in Assumption D.4, we require that all entries of $\boldsymbol{W}$ are initialized around a relatively small value $\omega$, which indicates that there is no huge difference between the magnitude of the initialization of major entries and minor entries.

However, notice that $\boldsymbol{W} = \boldsymbol{U}\boldsymbol{U}^\top$ and thus $w_{i,j} = \langle \boldsymbol{u} - i, \boldsymbol{u}_j \rangle$, where $\boldsymbol{u}_i \in \mathbb{R}^{d'}$ is the $i$-th row of $\boldsymbol{U}$. If we use Gaussian distribution to initialize $\boldsymbol{U}$, i.e. $\boldsymbol{u}_i \sim \mathcal{N}\left(\boldsymbol{0}, \tau^2 \boldsymbol{I}\right)$, where $\tau$ is a small real number, then we have the expectation of $w_{i,j}$ be

$$\mathbb{E} w_{i,j} = \begin{cases} 0 & i \neq j \\ d'\tau^2 & i = j, \end{cases} \tag{E.1}$$

which highlights the different between major entries and minor entries in initialization when $d'$ is large (and notice that $d'$ is lower bounded by $d$, which is the dataset dimensionality). Moreover, when $d'$ is small, the variance of $w_{i,j}$ will be large, so there is a greater chance for them to be away from 0.

### E.3 FAILURE MODES

A breakdown in the assumptions in App. D.1 can also lead to the model converging to "wrong" solutions that do not fully generalize OOD. For example, if a minor entry happens to be initialized too large (breaking the Assumption D.4), and / or the corresponding signal strength distinction is not large enough (breaking the Assumption D.5), then it is possible that the minor entry is not suppressed until it grows to a significant value, which can, in turn, lead to a too strong suppression on the corresponding major entry. In this case, a major entry might be suppressed to 0 (or at least, leave the initial phase from below) before it starts to grow, and thus never has chance to grow. This case corresponds to the model being "trapped" in a state that it only learns to compositionally generalize to a combination of certain (but not all) concepts.

In Fig. 10, we exhibit a case of such failure mode where the model fails to fully achieve OOD generalization. Notice how the loss value converges to a non-zero value and the major entry $w_{1,1}$ is suppressed at the very beginning and never grows. Additionally, the output trajectory is trapped at a point that combines only two directions, missing the third direction.

### E.4 FUTURE DIRECTIONS

We note that, current characterization of the model learning dynamics relies on the critical assumptions in App. D.1. Although those assumptions are reasonable and common in practice, the model

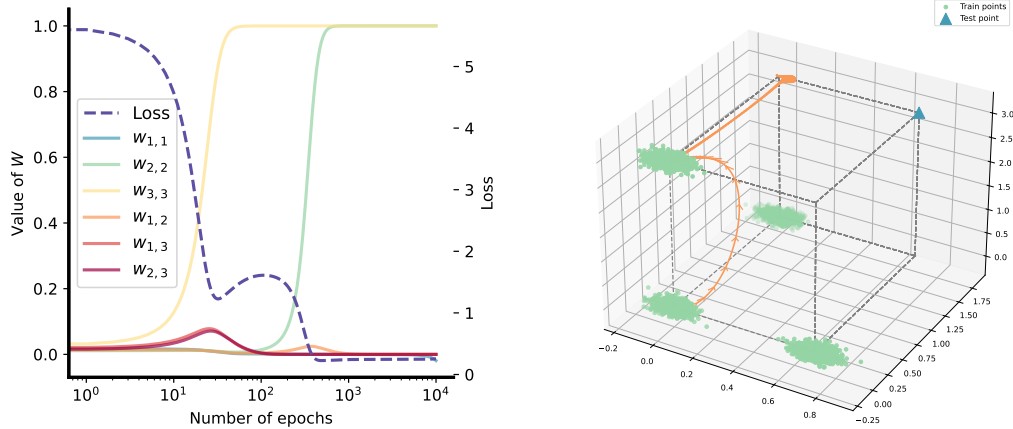

Figure 10: **An illustration of a failure case where the model doesn't successfully generalize OOD.** The dataset has $d = s = 3$, $\boldsymbol{\mu} = (0.7, 1.7, 3)$ and $\boldsymbol{\sigma} = (0.05, 0.05, 0.05)$. Left: The evolution of values of Jacobian and the OOD loss evaluated at test point $\hat{\boldsymbol{x}}$; Right: The output trajectory (orange curve).

behavior still shows some regularity when those assumptions breakdown. We have discussed some of the possible consequences when one of those assumptions breakdown above, but more in an intuitive way, instead of a systematic way. Therefore, one important future direction is to systematically characterize what will happen beyond the assumptions given in App. D.1. Among which, one specific and very important topic is the failure modes, i.e. under what conditions the model fails to generalize OOD.

Another important direction is to generalize current analysis to more complex models, such as deep linear networks, two-layer ReLU networks, or models in the NTK regime. The key point of current analysis of the 2-layer symmetric model is to correctly slice the learning dynamics of each entry of the Jacobian into multiple stages, such that in each stage, the learning dynamics is dominated by a rather simple dynamics.

Currently, since the model is 2-layer and without bias terms, there are only first-order and second-order terms in the learning dynamics. However, if we consider deeper models, there might be higher-order terms in the dynamics, and it is important to identify and simplify the effect these higher-order interactions in order to make the problem tractable.

For ReLU networks, it is known that there will be an "early-alignment" stage when trained on linear-separable data (Maennel et al., 2018; Min et al., 2023), where each neuron converge to a fixed direction, and make the model behaves like a linear model. We claim that investigating the early-alignment of 2-layer ReLU networks on the SIM task can be the starting point of theoretically characterizing the behavior of ReLU networks on the SIM task.

## F  ADDITIONAL SIM EXPERIMENT DETAILS AND RESULTS

In this section, we present the results of SIM experiments under different settings, including linear and non-linear models. The consistent behavior observed across these settings confirms the universality of our findings and explanations.

### F.1  EXPERIMENT DETAILS

In all SIM experiments, including those presented in main paper and in appendix, the number of training samples in each Gaussian cluster is $5000$. We use MLP models with either linear activations or ReLU activations, and all the models are trained using stochastic gradient descent with a batch size of $128$ and a learning rate of $0.1$ for $40$ epochs. Unless otherwise specified, the dimensionality of all data points is $d = 64$, and the hidden layer dimensionality of the models is also $64$ by default.

It is important to note that in our theory, we assumed that all training clusters and the test point are aligned with the standard coordinate. However, in our experiments, in order to make the results more universal and general, we add a random rotation to all the train / test points.

## F.2 ADDITIONAL EXPERIMENT RESULTS

Fig. 11 and Fig. 12 repeat the learning order experiments described in Sec. 3.1, using a 2-layer model with and without ReLU activation, respectively. It is easy to see that despite showing more non-regular curves, in multi-layer models the overall trends described in Sec. 3.1 and Sec. 3.2 are preserved.

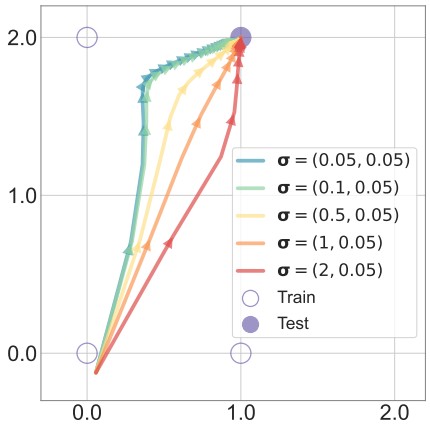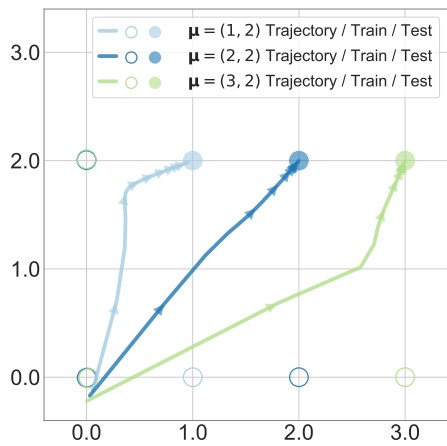

Figure 11: **Output trajectory of 2-layer models with linear activations**. The number of informative directions in the dataset is $s = 2$. Left: $\boldsymbol{\mu}_{:2} = (1, 2)$ with varied $\boldsymbol{\sigma}$'s; Right: $\boldsymbol{\sigma}_{:2} = (0.05, 0.05)$ with varied $\boldsymbol{\mu}$'s.

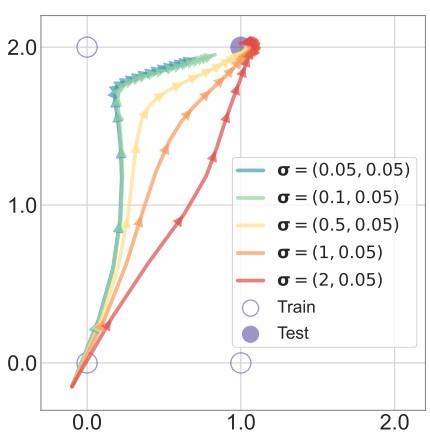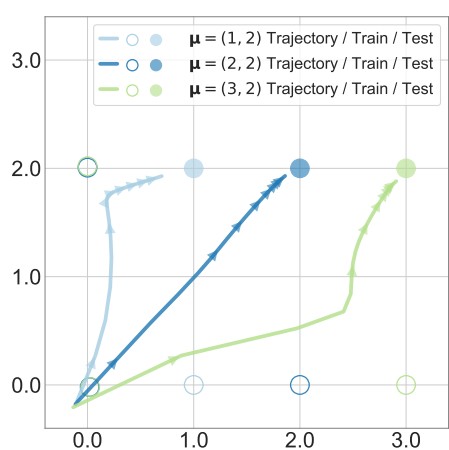

Figure 12: **Output trajectory of 2-layer models with ReLU activations**. The number of informative directions in the dataset is $s = 2$. Left: $\boldsymbol{\mu}_{:2} = (1, 2)$ with varied $\boldsymbol{\sigma}$ values; Right: $\boldsymbol{\sigma}_{:2} = (0.05, 0.05)$ with varied $\boldsymbol{\mu}$ values.

In Fig. 14, we present the output trajectory for two settings that exhibit significant Swing-by and Fig. 14 the corresponding loss curve. Specifically, the dataset has a dimensionality of $d = 3$, and is not randomly rotated. The models used have 3 layers, 3 hidden dimensions and linear activations. Comparing these results with the curve presented in Fig. 11, Fig. 12 and in Fig. 2, it is evident that models with more layers and fewer input dimensions are easier to have Swing-by, which confirms our theoretical prediction in Sec. 4.2.2.

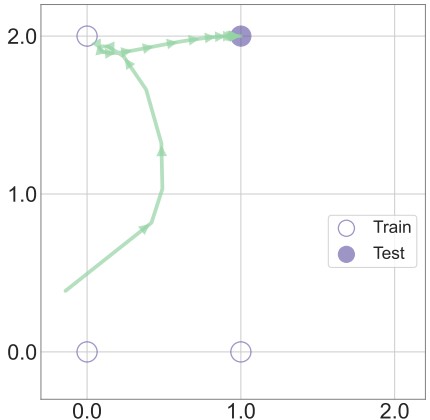 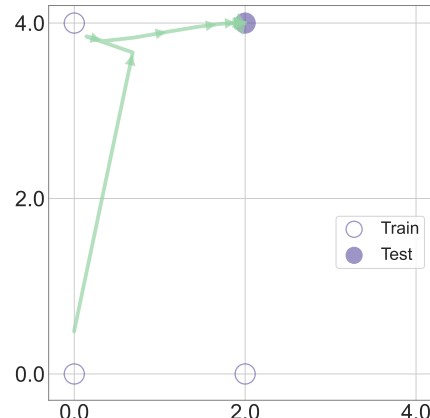

Figure 13: **Output trajectory of 3-layer models with linear activations and** $3$ **hidden dimensions.** The dataset has dimensionality $d = 3$, number of informative directions $s = 2$ and variance $\boldsymbol{\sigma}_{:2} = (0.05, 0.05)$. Left: $\boldsymbol{\mu}_{:2} = (1, 2)$; Right: $\boldsymbol{\mu}_{:2} = (2, 4)$.

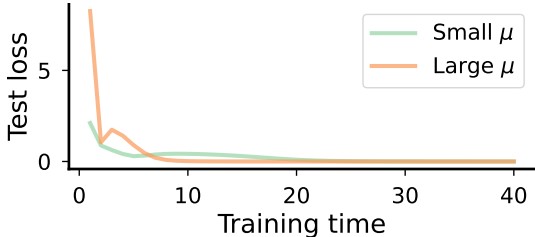

Figure 14: **The loss curve of corresponding models in Fig. 13.** Small $\mu$: $\boldsymbol{\mu}_{:2} = (1, 2)$; Large $\mu$: $\boldsymbol{\mu}_{:2} = (2, 4)$.

## F.3    FURTHER EXPERIMENTS ON THE ROLE OF OUT-OF-DISTRIBUTION

In Sec. 4.2.2, we briefly discussed that Swing-by must be an OOD phenomenon: the test point should be sufficiently away from the training clusters. In this subsection, we verify that this condition is satisfied in our SIM experiments.

In Fig. 15, we repeat the experiment of Fig. 2 (c), but in addition to the output curve, we also plot each training point. It is clear that there is no training point that is close to the test point.

In order to further ensure the separation of the training clusters with the testing point, in Fig. 16, we again repeat the experiment of Fig. 2 (c), but we make a truncation of the training distribution: we force that no training point can lie within the distance of $\frac{1}{2} \min_{p \in [s]} \mu_p$ of the testing point[5]. It is evident that only a few training samples are discarded and this truncation has little impact on the dynamics.

## F.4    FURTHER EXPERIMENTS ON THE TRAJECTORY OF TRAINING SET

In order to make it clearer to compare the model behavior on training set and test point, here in Fig. 17 and Fig. 18 we repeat the setting of Fig. 5, with additional information provided: 1) in Fig. 17, we added the training loss curve ; 2) In Fig. 18 we added two trajectories, corresponding to the model output given the training cluster means as input. We omit the training cluster centered at the origin since the 2-layer linear model always output $\mathbf{0}$ for this point.

---

[5]This a repeatedly sampling training points and discarding those within the specified distance of the testing point, until the training set reaches the desired size.

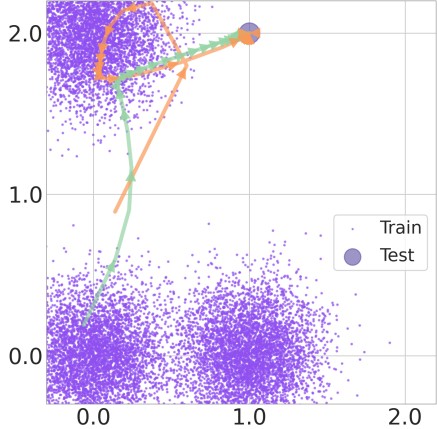

Figure 15: Reproducing Fig. 2 (c), with training points plotted.

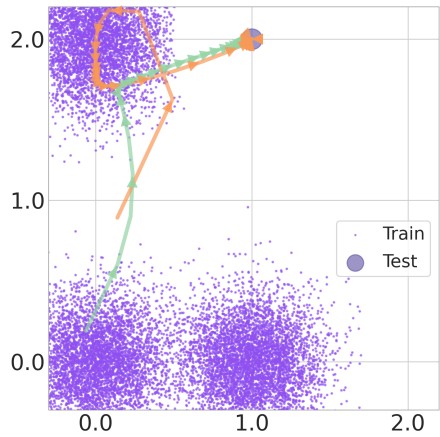

Figure 16: Reproducing Fig. 2 (c), with training distribution truncated.

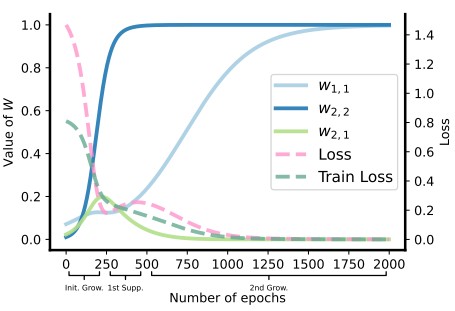

Figure 17: Reproducing Fig. 5 left, with training loss plotted.

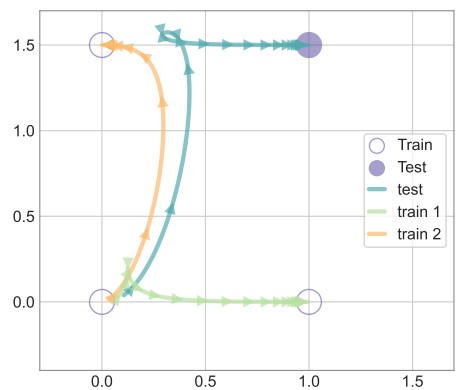

Figure 18: Reproducing Fig. 5 right, with the output trajectory of training set.

As shown in Fig. 17, the training loss monotonically decreases, exhibiting a different behavior compared to the non-monotonic test loss curve. This is expected because we used a small learning rate, and it is well-known that, under such conditions, the training loss must decay monotonically. In Fig. 18, the two curves of training trajectories both exhibit non-regular behaviors, but at different stages of the training. This observation aligns with our analysis of the multi-stage behavior of the learning.

## G   DIFFUSION MODEL EXPERIMENTS

We describe experimental details for the diffusion model experiments. We largely follow Park et al. (2024) in these experiments.

### G.1   SYNTHETIC DATA

Fig. 19 illustrates the DGP. We borrow part of the compositional data generating process (DGP) introduced by in Park et al. (2024). The DGP generates a set of images of circles based on the *concept variables* color={red,blue} and size={big,small}. Each concept variable can be selected as composed to yield four classes 00, 01, 10, 11 respectively corresponding to (red, big), (red, small), (blue, big), (blue, small). Here, class average pixels values of red

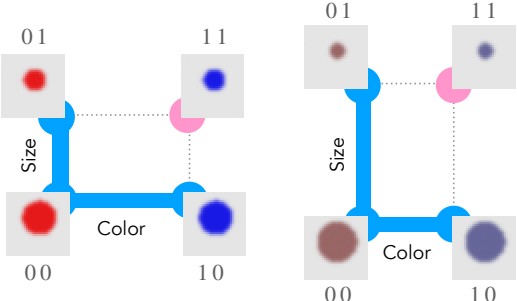

Figure 19: **Data Generation process with different concept signals.** Figure from (Park et al., 2024). The data generating processed used to train the diffusion model, where we can control the strength of the two concept signals independently. Left: A data distribution with a stronger concept signal in the `color` dimension. Right: A data distribution with a stronger concept signal in `size`.

and `blue` colors will control the concept signal for `color` and the difference between `small` and `big`. In Fig. 6, we fix the big circle's diameter to $70\%$ of the image and the small circle's diameter to $30\%$ of the image. We then adjust the absolute difference between the blue color and red color from 0.2 (very similar colors) to 0.7 (very different colors). The DGP randomizes the location of the circle, the background color and adds some noise to avoid having a very narrow data distribution. Please refer to Park et al. (2024) for further detail.

In Fig. 19, we show two different data distributions, one with a big color concept signal and one with a big size concept signal.

### G.2 MODEL & TRAINING

We train a conditional diffusion model on the synthetic data defined above. In specific, we train a variational diffusion model (Kingma et al., 2021) to generate $3 \times 32 \times 32$ images conditioned on a 4-dimensional vector where the first element of the vector specifies the size of the circle and the 3 others specifies the RGB colors.

**Model Architecture** We use a conditional U-Net (Ronneberger et al., 2015) with hidden dimensions $[64, 128, 256]$ before each downsampling layer and two ResNet (He et al., 2015) layers in each level. The conditioning vector is first transformed into the same dimensions as the hidden dimensions using a 2-layer MLP and are added to the representation after every downsampling layer. The U-Net has a self attention layer (Dosovitskiy et al., 2021) in its bottleneck. We used LayerNorm (Ba et al., 2016) for normalization layers and GELU (Hendrycks & Gimpel, 2023) activations.

**Diffusion** We use a learned linear noise schedule for the diffusion process as defined in Kingma et al. (2021), initialized with $\gamma_{\max} = 10$, $\gamma_{\min} = -5$. We assume a data noise of $1 \times 10^{-3}$. Variational diffusion models do not require fixing the number of diffusion steps at training time, but we use 100 steps for generation at inference time.

**Training** We train our model with the AdamW optimizer (Loshchilov & Hutter, 2019) with learning rate $1 \times 10^{-3}$ and weight decay 0.01. We use a batch size of 128 and train for 20k steps.

### G.3 EVALUATION

We evaluate the concept space representation of the generated output image using a trained classifier. Since we have the ground truth DGP, we used a large amount of data to train a perfect classifier. We used a U-Net backbone followed by a max pooling layer and a MLP classifier to classify each concept variable `color` and `size`. We train this classifier for 10k steps and achieve a $100\%$ accuracy on a held out test set. We average over 32 generated images and 5 model run seeds to get the ensemble average concept space representation.

The concept space MSE in Fig. 6 (b) is simply calculated as the MSE distance in the concept space defined in Park et al. (2024). The concept learning speed $|\mathrm{d}C/\mathrm{d}t|$ is quantified by estimating the movement speed in the same concept space by a finite difference method.

