# OpenReview forum: "Swing-by Dynamics in Concept Learning and Compositional Generalization"
_ICLR.cc/2025/Conference — ICLR 2025 Poster_

### Official Review · Reviewer_c6mH · 2024-10-28

**Soundness:** 3
**Presentation:** 3
**Contribution:** 3
**Rating:** 6
**Confidence:** 4

**Summary:**

This paper presents a theoretical framework for understanding the dynamics of compositional generalization in neural networks. Building on prior work that examines how models generalize by manipulating underlying primitive concepts, the authors introduce a Structured Identity Mapping (SIM) task, where a model learns the identity mapping on a Gaussian mixture with structured centroids. Through analysis of this SIM task, the paper reveals mechanisms behind non-monotonic learning dynamics, which is validated through experiments on text-conditioned diffusion models.

**Strengths:**

1. The paper’s explanation of learning dynamics, including the multi-stage Jacobian evolution and its impact on concept learning order and speed, sheds light on underlying model behaviors. I think the interesting point is the high similarity between the diffusion experiment and the theoretical model behavior, even though the correlation between the two is not clearly demonstrated.

2. Experiments on SIM and diffusion models are well-designed, with relevant results that support the theoretical predictions, enhancing the practical relevance of the results.

**Weaknesses:**

1. The theoretical model studied in this paper differs significantly from practical diffusion models. Although the authors attempt to demonstrate a relationship between the theoretical analysis and practical model behavior through empirical results, this connection remains unclear, making it challenging to confirm whether the behavior observed in diffusion models is directly related to the theoretical insights presented. While I acknowledge the value of the analysis for single-layer and two-layer linear networks, it would be beneficial if the authors conducted further analysis on simplified diffusion settings to bridge the gap between theory and practice more effectively.

2. In reference [1], the authors also discuss learning primitives in compositional tasks. Additional discussion on this work could be beneficial. Based on this, I have the following question: In [1], small initialization is an important factor for the model to learn primitives. In this paper’s theoretical part, small initialization is also assumed, but in practical diffusion models, is small initialization a crucial factor for learning primitives, or is it merely a theoretical necessity?

3. Could Figure 5 display the training data’s trajectory evolution? I am curious whether there is a similarity between the training and test trajectories in the early stages of training.

[1] Initialization is Critical to Whether Transformers Fit Composite Functions by Reasoning or Memorizing. arXiv:2405.05409.

**Questions:**

See weaknesses.

---

> ### Author Response · Authors · 2024-11-24
> **Response to Reviewer c6mH**
>
> We thank the reviewer for the insightful feedback. Below, we address each concern raised by the reviewer.
>
> **The theoretical model studied in this paper differs significantly from practical diffusion models**
>
>
> We thank the reviewer for recognizing the value of our theoretical analysis. We believe that the SIM task is as a reasonable proxy for studying compositional generalization, as it successfully captures and predicts many phenomena previously   observed in compositional generalization. We kindly refer the reviewer to the General Response for a more detailed discussion of this point.
>
> **is small initialization a crucial factor for learning primitives, or is it merely a theoretical necessity?**
>
> Previous work has reported that small initialization can sometimes be critical for generalization (e.g., [1], [2]). While we are not aware of any public results directly addressing the effect of small initialization on the empirical performance of compositional generalization, our own experience suggests that small initialization plays a significant role in improving compositional generalization performance.
>
>
> **Could Figure 5 display the training data’s trajectory evolution? I am curious whether there is a similarity between the training and test trajectories in the early stages of training.**
>
> We have revised the paper to include an additional section (Appendix F.4) presenting the training loss and the output trajectory of the model when given training points as input, under the same settings as Figure 5. From the additional results, it can be observed that:
> 1. The training loss monotonically decreases, contrasting with the non-monotonic test loss curve.
> 2. The training trajectory exhibits stage-wise non-regular behaviors. These behaviors result from the stage-wise evolution of the Jacobian, aligning with our theoretical analysis.
>
> **Summary**
>
> We appreciate the reviewer’s thoughtful feedback, which has helped refine and enhance our paper. We hope our answers and the updates made in the revision can help address your concerns.
>
> &nbsp;
>
> [1] Omnigrok: Grokking Beyond Algorithmic Data
>
> [2] Training behavior of deep neural network in frequency domain

---

> > ### Comment · Reviewer_c6mH · 2024-11-26
> >
> > Thank you for your detailed responses and the additional clarifications. I appreciate the effort you have put into addressing my earlier concerns. However, I still have two key reservations:
> >
> > **Initialization and OOD Generalization:**
> > In [1], the authors demonstrate that small initialization plays a critical role in enabling transformers to learn primitive operations in composite tasks, leading to improved OOD generalization. Conversely, larger initializations fail to achieve similar results. This seems to contradict aspects of your response, particularly regarding "While we are not aware of any public results directly addressing the effect of small initialization on the empirical performance of compositional generalization". Given the reliance on small initialization in your theoretical proof, I believe it would be beneficial to discuss related literature explicitly, such as [1, 2, 3].
> >
> > **Theoretical Model and Compositional Generalization:**
> > Your title emphasizes compositional generalization, but I am curious about the implications of the theoretical framework adopted in the paper, particularly the two-layer linear network architecture. Specifically, does your theory suggest that as long as the training set fits perfectly, the model inherently generalizes to unseen composite tasks? If so, I would appreciate further elaboration or clarity regarding the connection between this theoretical property and practical compositional generalization in more complex, nonlinear networks.
> >
> > [1] Initialization is Critical to Whether Transformers Fit Composite Functions by Reasoning or Memorizing
> >
> > [2] Omnigrok: Grokking Beyond Algorithmic Data
> >
> > [3] Training Behavior of Deep Neural Networks in the Frequency Domain

---

> ### Author Response · Authors · 2024-11-27
> **Further Response to Reviewer c6mH**
>
> Thank you so much for your thoughtful feedback. We would now like to address your two remaining concerns.
>
> **Initialization and OOD Generalization**
>
> Thank you so much for bringing these insightful references to our attention. We have carefully reviewed these works, and were excited to learn about especially [1], which studied the role of initialization on OOD generalization in compositional tasks, exactly the reference we were looking for! We have now reflected your feedback by updating our manuscript to incorporate discussions of [1,2,3] in both Section 4.2.1 (line: 398) and Appendix D.3 (line 1022).
>
>
> **Theoretical Model and Compositional Generalization**
>
> Thank you for this valuable question. The previous work on diffusion models [4, 5] that we build upon first demonstrated that diffusion models can compositionally generalize, then focused on the phenomenology of learning dynamics - specifically, the process by which the model learn individual concepts to achieve compositional generalization. This motivated our theoretical analysis to focus on the **learning dynamics** of compositional generalization rather than the question of whether generalization occurs. While our empirical results on the SIM task in Section 3 include deep non-linear models reproducing rich phenomenology of diffusion models, our theoretical analysis uses a deep linear model, which inherently guarantees compositional generalization once all training data is perfectly fit.
>
> Regarding the theory of non-linear models, we have now discussed in Appendix E.4, Future Directions section. Specifically, for ReLU networks, in our preliminary experiments we have observed an extra "early alignment" phase around initialization, in which the Jacobian of every point in the concept space converges to roughly the same value, and then the model behavior will be similar to a deep linear model, and our theoretical analysis in this paper applies. This "early alignment" phase of deep ReLU networks has been proved in previous work for 2-layer ReLU networks under linear-separable data and exponential classification loss. Since we have observed it in the SIM setting, one of the important future directions is to prove it for the 2-layer ReLU network on the SIM task, and thus expand our theoretical analysis to the non-linear setting.
>
> Thank you for actively engaging during the rebuttal period and providing insightful feedback. We hope this addresses your remaining concerns.
>
> [1] Initialization is Critical to Whether Transformers Fit Composite Functions by Reasoning or Memorizing
>
> [2] Omnigrok: Grokking Beyond Algorithmic Data
>
> [3] Training Behavior of Deep Neural Networks in the Frequency Domain
>
> [4] Compositional Abilities Emerge Multiplicatively: Exploring Diffusion Models on a Synthetic Task
>
> [5] Emergence of Hidden Capabilities: Exploring Learning Dynamics in Concept Space

---

> ### Comment · Reviewer_c6mH · 2024-12-03
>
> Thank you for your thoughtful response and for addressing the concerns raised. I appreciate the updates to your manuscript.
>
> I fully acknowledge and agree with your understanding of early alignment in non-linear networks. Indeed, I believe that proving early alignment in non-linear networks is a promising direction and an attainable goal. My previous suggestion to examine the training trajectories of both training and test data was driven by an interest in observing such early alignment phenomena, which I think is a crucial aspect of your work.
>
> Overall, I believe the contributions of your paper are solid and relevant. I remain inclined to accept the paper, and I once again thank you for your thoughtful rebuttal and for addressing my concerns.

---

### Official Review · Reviewer_pvo7 · 2024-11-04

**Soundness:** 3
**Presentation:** 3
**Contribution:** 3
**Rating:** 6
**Confidence:** 3

**Summary:**

The paper introduces the structured identity mapping (SIM) task, where a network is trained to learn the identity mapping from inputs which are sampled from largely disjoint Gaussian distributions, and evaluated on a test point defined as the average of their means. By analyzing learning dynamics of one-layer and symmetric two-layer linear networks on this task, the paper shows that several existing empirical observations can be modeled - higher SNR features are learnt earlier and faster, an epoch-wise double-descent phenomenon (termed Transient Memorization) occurs on the test point, and that learning slows down exponentially with time.

**Strengths:**

- Paper formalizes a proxy task (SIM) to model the learning dynamics of compositional generalization, and shows that real-world text-conditioned diffusion models exhibit similar behavior on a specific task.
- Theoretical setup is clearly explained and theoretical conclusions are also well-elaborated. Limitations of theoretical results on the one-layer and symmetric two-layer linear models are also clearly discussed.

**Weaknesses:**

- It is not at all apparent that $\hat{x}$ is "outside of the training distribution", since $p(x_k^{(p)} = \hat{x}) > 0$ for training data $x_k^{(p)}$, despite what is claimed on L174. Why not choose sigma such that $\sigma_p \geq 0$ while keeping $\sigma_{q \neq p} = 0$? Furthermore components of $\sigma$ seems to be as large as $2$ in Fig 1., with $\mu$ ranging from $0 - 2$. In such cases, the converse seems to hold -- that $x_k^{(p)}$ is very much in-distribution of the training data.
- As a result, the introduced terminology "Transient Memorization" does not seem to be different from epoch-wise double descent. The paper stresses that the main difference is OOD vs in-distribution test loss (L263-265), but (a) as mentioned above the test sample considered are in fact in-distribution, and (b) even for double descent, it is actually essential for testing samples to differ significantly from the training ones as pointed out by [1].
-  The proposed SIM task is claimed to be a "further abstraction of the 'concept space' previously explored" (L531), however it appears instead to be an (over)-simplification of previous investigations. The motivation of the SIM model and how it relates with more general real-world settings are also unclear (for instance, why the proposed auto-encoding loss?).
- The paper's main empirical results section (Page 10) appear very underdeveloped and all important experiment details are left to the Appendix. Figure 6 is also not clearly explained, and different plots are distinguished based on "difference of class mean pixel values". Without reading App G, it is almost impossible to interpret what these results and experiments are doing. $\Delta$ Color is not defined in App G either.
- Paper concludes from Fig. 2 that "deceleration is not determined by ... the loss value ... but more depends on the data and training time" (L215). This is not self-evident, since (1) the training loss is not even plotted in Fig 2, and (2) regions with denser arrows are clearly also regions of lower test loss.
- The paper also claims that "there is a timescale determined by the training data such that if the model does not achieve OOD generalization within that period, significantly more computation will be required for the model to achieve it." (L238-240), which purportedly can be observed in Fig 2(b). However, the figure simply shows that configurations with lower $\sigma$ values take longer to correctly predict the test point. This conclusion seems self-fulfilling, since when $\sigma$ is larger, the density of training samples around $\hat{x}$ will be much higher.
- Minor: L534-L535 incomplete sentence - "We make a comprehensive."
- The work's primary area is stated as interpretability / explainability in AI, but the contributions to these areas are unclear. The bulk of the theoretical contributions seem to be instead exploring learning dynamics for one-layer/deep-linear models, which I have insufficient experience to evaluate with regards to recent literature.

[1]  Double Descent Demystified: Identifying, Interpreting & Ablating the Sources of a Deep Learning Puzzle, 2023.

**Questions:**

See weaknesses

---

> ### Author Response · Authors · 2024-11-24
> **Response to Reviewer pvo7**
>
> We thank the reviewer for the detailed and insightful review. The reviewer has raised concerns regarding the OOD setting, Transient Memorization, and the validity of SIM as an abstraction of compositional generalization. These are indeed important issues that have been mentioned by multiple reviewers; therefore, we kindly refer the reviewer to the General Response, where these concerns are discussed and addressed in detail.
>
> Additionally, we appreciate the reviewer’s feedback that Section 5 was unclear. In response, we have rewritten the entire section to improve clarity.
>
> Below, we address the remaining concerns raised by the reviewer.
>
> **the introduced terminology "Transient Memorization" does not seem to be different from epoch-wise double descent**
>
> We kindly refer the reviewer to the General Response for a more comprehensive discussion about Transient Memorization. Here, we would like to make a few comments regarding the relationship between Transient Memorization and epoch-wise double descent.
>
> It is indeed true that many papers, including [1] (as suggested by the reviewer) and [2] (which our paper cites), study epoch-wise double descent in settings where the training and test distributions differ significantly. However, we would like to make the following points:
> 1. In [1] and [2], the difference between training and test distribution arises from that training distribution is a noisy version of the test distribuition. However, the SIM task is fundamentally different from these settings, and in the settings where Transient Memorization happens we actually have the test point being very far from the training  distribution (again, see the General Response for further elaboration on this point).
> 2. The underlying mechanism of Transient Memorization also significantly different from [1] and [2]. Transient Memorization comes from the high-order interactions between different directions, rather than from training noise.
> 3. Moreover, Transient Memorization is not caused by any of the commonly known sources of non-monotonic test loss, such as large step size, noisy training, or over-parameterization, highlighting that this is a new mechanism leading to non-monotonic test loss curves (again, see General Response for more detailed discussions regarding this point).
> 4. Finally, it is okay to refer to the test loss curve of Transient Memorization as an epoch-wise double descent curve. Ultimately, this is only a matter of terminology.
>
> **Inappropriate claims about terminal time slowing down**
>
> We thank the reviewer for highlighting this issue. We agree that some descriptions of the terminal time slowing down were not sufficiently rigorous. In response, we have revised the paper to remove all unclear or confusing statements regarding the terminal phase slowing down.
>
>
> **Minor: L534-L535 incomplete sentence - "We make a comprehensive."**
>
> We thank the reviewer for pointing this out. We have deleted this incomplete sentence in our revision.
>
>
> **The work's primary area is stated as interpretability / explainability in AI, but the contributions to these areas are unclear. The bulk of the theoretical contributions seem to be instead exploring learning dynamics for one-layer/deep-linear models, which I have insufficient experience to evaluate with regards to recent literature.**
>
> We appreciate the reviewer’s feedback on this point. It is correct that the core focus of this paper lies in studying the learning dynamics of one-layer / deep-linear models. However, our goal is different from typical studies in learning dynamics or optimization, which often focus on proving convergence or identifying properties of the convergent point. In contrast, our objective in analyzing learning dynamics is to understand the mechanisms underlying previously observed phenomena in compositional generalization. By providing insights into these mechanisms, we believe our work aligns naturally with the primary area of interpretability/explainability in AI.
>
>
>
> **Summary**
>
> We sincerely thank the reviewer for the thoughtful feedback, which has helped improve the clarity and rigorousness of our paper. We hope our answers and the updates made in the revision can help address your concerns.
>
>  &nbsp;
>
>
> [1] Double Descent Demystified: Identifying, Interpreting & Ablating the Sources of a Deep Learning Puzzle
>
> [2] Towards Understanding Epoch-wise Double descent in Two-layer Linear Neural Networks

---

### Official Review · Reviewer_caEk · 2024-11-04

**Soundness:** 3
**Presentation:** 3
**Contribution:** 3
**Rating:** 6
**Confidence:** 4

**Summary:**

This paper investigates the dynamics of neural networks in achieving compositional generalization. The authors propose a Structured Identity Mapping (SIM) task, where models are trained on a Gaussian mixture dataset organized around concept-centric clusters. They find that concept strength and diversity strongly influence the speed of convergence and the ability to generalize to compositional out-of-distribution (OOD) test points. They also observe a phenomenon called "transient memorization," where models initially memorize the training distribution but eventually reorient towards OOD generalization with extended training. Theoretical analyses on simple linear models substantiate these findings, highlighting the role of signal strength and diversity in learning order and the occurrence of non-monotonic loss behavior. Finally, the authors validate these findings by training diffusion models, observing similar patterns in generalization dynamics.

**Strengths:**

1. The paper is clearly written, with a logical flow that makes each insight and conclusion easy to follow.
2. The simplicity of the problem setup enhances the clarity and robustness of both the empirical observations and the theoretical contributions.
3. The diffusion model results are compelling, mirroring the behavior observed in simpler settings and offering explanations for phenomena noted in prior work.

**Weaknesses:**

See the questions section for more information.

**Questions:**

The paper effectively accomplishes its aims, though a few points could be noted for improvement:
1. **Limited Theoretical Scope**: While the theoretical framework successfully supports the empirical observations in more complex models, it is based on simple linear models.  I acknowledge the difficulty in analyzing more complex neural networks theoretically. Nonetheless, for future work, extending the analysis to objects like kernel methods, particularly the Neural Tangent Kernel (NTK) [1], could provide a deeper understanding as NTK has been shown to capture certain behaviors of neural networks during training [2].
2. **Contextualization of Findings**: Some insights, such as the speed and order of generalization, have been observed in different forms in previous OOD works [3, 4, 5]. Further contextualizing the findings within these works could enhance the paper’s relevance and depth.

[1] [Jacot 2018] https://arxiv.org/abs/1806.07572

[2] [Lee 2020] https://arxiv.org/abs/2007.15801

[3] [Nagarajan 2020] https://arxiv.org/abs/2010.15775v3

[4] [Arjovsky 2019] https://arxiv.org/abs/1907.02893

[5] [Rosenfeld 2020] https://arxiv.org/abs/2010.05761

---

> ### Author Response · Authors · 2024-11-24
> **Response to Reviewer caEk**
>
> We thank the reviewer for acknowledging the quality of our writing, problem setup, and experiments. Below, we address each question and suggestion raised by the reviewer.
>
>
> **Limited Theoretical Scope: for future work, extending the analysis to objects like kernel methods, particularly the Neural Tangent Kernel (NTK), could provide a deeper understanding as NTK has been shown to capture certain behaviors of neural networks during training.**
>
> We agree that it is important in the future work to extend the current analysis to more complex models. In fact, in Appendix E.4 we have outlined a roadmap towards extending the current theoretical analysis to 2-layer ReLU networks, which will include tackling the early alignment phenomenon in 2-layer ReLU networks. We thank the reviewer for highlighting the NTK setting, and in response, we have revised Appendix E.4 to explicitly include NTK as a potential direction for future exploration.
>
>
> **Contextualization of Findings: Some insights, such as the speed and order of generalization, have been observed in different forms in previous OOD works [3, 4, 5]. Further contextualizing the findings within these works could enhance the paper’s relevance and depth.**
>
> We thank the reviewer for pointing out these relevant references. We have revised the paper to include these works as references and to better contextualize our findings within the broader literature.
>
>
> **Summary**
>
> Thank you again for your positive feedback regarding our writing, theoretical results, and empirical validation using diffusion models! To fully address your feedback, we have now revised the paper to expand the discussion about future work and improved the contextualization of our findings. Please don't hesitate to share any additional concerns that we should address.

---

### Official Review · Reviewer_DJMW · 2024-11-04

**Soundness:** 3
**Presentation:** 2
**Contribution:** 3
**Rating:** 8
**Confidence:** 4

**Summary:**

The paper sets out to explain previously reported characteristics of the learning dynamics of text-conditioned diffusion models with respect to  their ability to compose concepts to generalize to unseen combinations. Specifically, the paper aims to study why models learn concepts in a specific order and how the structure of the data influence a model's learning speed. To this end, the authors propose a simple reconstruction task (which they term structured identity mapping, SIM) in which an MLP should reconstruct its inputs drawn from a mixture of Gausssians. The authors argue that this task allows them to study and explain the key empirical observations on compositionally generalizing diffusion models and lead to the novel characterization of a non-monotonic learning dynamic.

**Strengths:**

This work studies a relevant problem: The learning dynamics of compositional generalization in diffusion models are important to understand how models can learn in a sample-efficient manner, how generalization can be achieved, or how training data should be curated, to name a few ways insights could be impactful.

The paper is easy to follow for the most parts and builds on a prior line of work in this area.

The non-monotonic training dynamics of a symmetric two-layer linear model are well explained by the theory and offer a novel (as far as I can tell) characterization of the training behavior of basic models. The theoretical training dynamics outlined in §4 and especially §4.2 and Fig. 5 model the empirical observations from §3 and Fig. 2 well.

**Weaknesses:**

In summary, I find the paper misses the mark, as the SIM task is, as far as I understand, a poor setting to study compositional generalization, and the insights on this simple task translate poorly to the training of a diffusion model, even for the simple toy setting that is used (which itself is approximating the compositional generalization of text-conditioned diffusion models that this paper aims to study). I will elucidate the issues I see with the setting and results below.

As it stands, I cannot recommend acceptance of this work; however, I could see how the results on the learning dynamics of symmetric linear models from §4 might be interesting in their own right, without considering any (as I understand, faulty) connection to compositional generalization. Maybe a reconsideration of the scope of the paper and a reframing of the results without overclaiming meaningful insights regarding the training dynamics of text-conditioned diffusion models could be an interseting contribution on its own.

# The SIM Setting
As far as I understand, previous works took the "concept space" as an abstracted view of the training and test data in order to study training trajectories of diffusion models. In the SIM task, this concept space is instead interpreted directly as the input and output space of a model. While I can see that this choice is motivated by the intuition in LL89, "a good generator essentially performs as an identity mapping in the concept space", this simplification of the setting is not explicitly mentioned, e.g., in the introduction, and the limitations this simplification imposes on the applicability and transferability of the results are not discussed.

Further, the training distribution in the SIM task is a mixture of Gaussians. In general, I find that this setup is not explained very clearly, see minor issues below. The central issue I see is that this setup entirely misses the point of (compositional) generalization. First, by introducing the training clusters as Gaussians, any point in $\mathbb R^d$ has probability $> 0$. So it is simply not true that any "[test] point is outside of the training distribution—not just the training data, necessitating out-of-distribution generalization." (LL173). Second, since test loss is only tracked on an individual point, not a test distribution, the experiments cannot even consider distribution shifts. Overall, describing this setting as requiring "generalization" to an "OOD test point" is misleading.

If we understand that in this setting, any test point has non-zero probablity in the training set, the conclusions from §3 are mostly unsurprising. Consider the probability density of a given point in the grids in Fig. 2 belonging to the training set (in fact, I encourage the authors to include this information in the figure). I expect that for Fig. 2a, we will see that the trajectories are skewed towards higher-density regions, which roughly speaking can be interpreted as the model reducing the risk of a certain output. Similarly, we can understand the takeaway from Fig. 2b, the "terminal slow down" differing for different values of $\boldsymbol \sigma$ as an effect of the probability density. For $\boldsymbol \sigma = (2, 0.05)$, the overall density at point $(1, 2)$ is much higher than for $\boldsymbol \sigma = (0.05, 0.05)$, making it much less likely that this point (or points close to it) are sampled, so that the model takes much longer to learn it.

While the "transient memorization" is an intersting behavior of even simple models, and the author's explanation of this phenomenon seems interesting, I find it hard to justify translating any insights from this setting to a compositional generalization regime.

# Transferring Results to Diffusion Models
§5 is very bare-bones, to the point that it is unclear how well the observations from the SIM setting actually translate here. E.g., there doesn't seem to be as much of a "terminal slow down" in Fig. 6 as was shown in Fig. 2a/b. The "transient memorization" also seems much less pronounced.

Additionally, it is unclear _why_ results form the SIM setting should translate here, as in this case the test point is truly out of distribution, and, if I understand the training setting correctly, the trianing set only contains discrete values for each factor.

**Questions:**

# Questions
- LL86: "The model output is then passed through a classifier which produces a vector indicating how accurately the corresponding concepts are generated (e.g. a generated image of big blue triangle might be classified as (0.8, 0.1, 0.9)). In this way, the process of generation becomes a vector mapping, and a good generator essentially performs as an identity mapping in the concept space." In this setup, the vector mapping is $c \circ g$, comprising the generator $g$ _and_ classifier $c$. While it is clear that a good generator should be the identity, the role of the classifier also has to be analyzed. Can we be sure a priori, that $c$ is an identity mapping?
- How is training done? Is a fixed number of points sampled in the beginning to be used as a training set, or are points instead drawn for each batch?
- LL193: Why is $\mu_k$ equated with signal strength? Intuitively, instead of the absolute value of $\mu_k$, the distance between clusters should be more meaningful, which might be high for an individual cluster even if $\mu_k$ is low.
    - it is also not clear why this should matter to the model. The model could simply normalize each input dimension such that inputs are always balanced, e.g., in the task of Fig. 2b
- App. B: This observation mainly seems to imply that test loss _increases with the distance from the training set_, which, again does not say anything about OOD generalization or compositional generalization, and instead is a clear effect of the increasing probability density of the training set.
    - Also, why is the loss truncated instead of normalized?

# Minor Suggestions
- Fig. 1 and LL73 show that the data is clustered around nodes of the hypercube, yet LL150 and LL160 explain that each cluster has a different $\mu_p$ (distance from the origin). How do these statements fit together?
- L163 “There is also optionally a cluster centered at 0.” Is this in addition to the $s$ clusters, or is this just specifying that $\mu_p$ might be 0?
- LL170: "We evaluate the model at a Gaussian cluster centered at the point that combines the cluster means of all training clusters." What does this mean? Say the test cluster is $\boldsymbol x_k^\text{test} \sim \mathcal N(\boldsymbol \mu_\text{test}, \boldsymbol \sigma_\text{test}^2)$, does this just mean that $\boldsymbol \mu_\text{test}$ is the mean of all $\mu_p \boldsymbol 1_p$?
- §3.1 and Fig. 2: What are the markers at each optimization timepoint? The model output for the given input $\boldsymbol x = \boldsymbol \mu$?

---

> ### Author Response · Authors · 2024-11-24
> **Response to Reviewer DJMW (Part 1)**
>
> We appreciate the reviewer's detailed comments; we are pleased to see that the reviewer finds our paper  easy to follow and our theoretical results are interesting and novel. The primary concern raised by the reviewer pertains to the validity of the SIM task as a proxy for exploring compositional generalization and the unclear connection between the SIM task and the diffusion model experiments. These are indeed two critical issues that we would like to address. We kindly refer the reviewer to the General Response for a related discussion of these concerns. Below, we make additional clarifications and answer to the questions and minor suggestions.
>
>
> ## The SIM Setting models Out-of-Distribution Senarios
>
> > First, by introducing the training clusters as Gaussians, any point in R^d has probability > 0. So it is simply not true that any "[test] point is outside of the training distribution—not just the training data, necessitating out-of-distribution generalization." (LL173).
>
> > Second, since test loss is only tracked on an individual point, not a test distribution, the experiments cannot even consider distribution shifts.
>
> Thank you for these important points of clarification. While it is technically correct that any point in $\mathbb R^d$ has non-zero probability under a Gaussian mixture, the Gaussian distribution exponentially decays with distance from the centroids. Our study specifically focuses on parameter regimes of Gaussians where the test point $\widehat{\mathbf{x}}$ lies far away from training data points, in regions where the training distribution density is exponentially small (technically non-zero but negligible for practical purposes). The only exception is some selected curves in Figure 2(b); those are to demonstrate an edge case where an extremely large and highly imbalanced $\sigma$ can reverse the direction of the learning dynamics. and should not be regarded as a general configuration of the SIM task. In fact, **our main theoretical results inheritly requires very small $\sigma$, i.e. practically "zero" overlap between the test point and training points**. We have added a Remark in Section 4.2.2 to make these requirements explicit.
>
> Furthermore, we have **added new experiments reproducing our results with strictly truncated Gaussians that ensure zero training point in the quadrant region containing the test point, i.e., strictly out of distribution scenario (Figure 16)**. These experiments confirm that our key findings about learning dynamics hold even under this stricter OOD condition. This setup allows us to study meaningful out-of-distribution generalization, analogous to how real-world compositional generalization often requires extrapolating to regions far from the training data.
>
> > ***If we understand that in this setting, any test point has non-zero probability in the training set***, the conclusions from §3 are mostly unsurprising...
>
> As we discussed and clarified above, **we primarily focus on cases where any test point has vanishing, or practically zero, probability in the training set**, except for edge cases presented in Figure 2\(b\). To fully address your concern, we have now:
>
> 1. **Minimal Overlap** \[Figure 1\(c\), Figure 15\]: As demonstrated in our new Figure 15, the Gaussian clusters in the training data practically have negligible overlap with test point, with practically zero density at the test point location. The variance of each cluster is specifically chosen to ensure this property throughout the paper.
>
> 2. **Added new experiments reproducing our results with strictly truncated Gaussians that ensure strictly zero overlap with the test point, i.e., strictly out of distribution scenario (Figure 16)**.
>
> These experiments reproduce all our key findings, confirming that our observations are not artifacts of theoretical non-zero probabilities in the tails of Gaussian distributions.
>
>
> ## Concerns about transferring results to diffusion models
>
> We thank the reviewer for pointing out that Section 5 was not sufficiently clear. In response, we have rewritten Section 5 to provide a clearer explanation of the experiments and their implications.
>
>
> We would like to emphasize that the terminal phase slowing down observed in Figure 6 (c) is actually significant. Since the y-axis is presented on a log scale and the curve exhibits a nearly constant negative slope, this indicates that the loss is decaying exponentially. This behavior aligns perfectly with the theoretical predictions.
>
> Additionally, in Figure 6(a), we observe clear patterns where the model’s output initially moves toward the training distribution before shifting toward the generalization direction. This behavior is consistent with our description of Transient Memorization.

---

> ### Author Response · Authors · 2024-11-24
> **Response to Reviewer DJMW (Part 2)**
>
> ## Responses to Questions
>
> Below, we address each question raised by the reviewer:
>
> **Question regarding LL86**
>
> The statement "a good generator essentially performs as an identity mapping in the concept space" is intended as an intuitive explanation rather than a precise one. Specifically, we mean that, ideally, $c \circ g$ should be an identity mapping. This follows naturally from the idea that classification should ideally reverse the process of generation.
>
>
>
> **Question regarding training**
> For both SIM experiments and diffusion experiments, we first sample a number of training samples and fix them, and then do full-batch training.
>
> **Question regarding LL193**
>
> The SIM task serves as an analogy for the concept space. In this context, it is natural for samples that have stronger concept signal to be farther from the origin.
>
>
>
> **Question regarding Appendix B**
>
> As noted in the General Response, the probability of any training point being close to the test point is very small.
>
> Regarding the choice of truncating the losses rather than normalizing them: this approach preserves the same scale across different epochs, making it easier to compare the loss values of each node across different epochs.
>
>
> ## Responses to Minor Suggestions
>
> **Regarding different $\mu$'s**
>
> Thank you for raising this point. Yes, since the edge lengths can differ, this shape is indeed a hyperrectangle rather than a hypercube to be priecise. We have updated the manuscript to reflect your feedback.
>
>
> **Regarding L163**
>
> Yes, it is in addition to the $s$ clusters, and we have now clarified this point in the updated manuscript to reflect your feedback.
>
> **Regarding L170**
>
> We respectfully confirm that the mathematical expression in the paper is correct: the test point $\widehat {\mathbf{x}} = \sum_{p=1}^s \mu_p {\mathbf{1}}_p$ is indeed a sum, not a mean, of the centroids (note there is no division by $s$). Figure 1\(c\) illustrates this explicitly: when the training centroids are at $\{(0,0,0), (\mu_1,0,0), (0,\mu_2,0), (0,0,\mu_3)\}$, their summation produces the test point $(\mu_1,\mu_2,\mu_3)$ (shown as the red dot). This construction is central to our experimental design, as it deliberately creates an out-of-distribution test point in most of our studied setups.
>
> **Regarding §3.1 and Figure 2**
>
> Each marker represents the model output at a specific time point, given the input  $\widehat {\mathbf{x}}$.
>
>
> ## Summary
> Thank you again for your valuable critical feedback, which helped us clarify the key implicit assumption for ensuring our SIM task respect the OOD nature. Through the clarifications made here, in the General Response, and in the manuscript revisions, we hope to have addressed the reviewer’s concerns thoroughly.

---

> > ### Author Response · Authors · 2024-11-25
> > **Thank you for increasing your score to 8!**
> >
> > Thank you for increasing your score to 8, now strongly supporting acceptance of our paper! We are grateful that our revisions demonstrated how Gaussian mixtures make SIM Task an effective model for compositional generalization, our theoretical results reproduce key diffusion model phenomena despite simplicity, and our novel prediction was validated empirically. We thank you again for your thoughtful and valuable feedback that has greatly helped clarify and contextualize our theory's contributions to understanding the challenge problem of compositional generalization in diffusion models.

---

### Author Response · Authors · 2024-11-24
**General Response (Part 1)**

We sincerely thank all reviewers for their thoughtful feedback. We are encouraged that reviewers found our paper to be "clearly written, with a logical flow" (Reviewer caEk); our "theoretical setup and conclusions are well-elaborated" (Reviewer pvo7); and our "experiments are well-designed" and "support theoretical predictions" (Reviewer c6mH). The reviewers also highlighted several important contributions: (1) our "novel characterization of the training behavior" of symmetric linear models (Reviewer DJMW), (2) our "formalization of a proxy task" for studying compositional generalization (Reviewer pvo7), and (3) our "compelling" diffusion model results that "mirror the behavior observed in simpler settings" (Reviewer caEk).

We are particularly encouraged that three reviewers recommend acceptance, and Reviewer DJMW, who currently recommends rejection, notes that our results on learning dynamics of symmetric linear models might be "interesting in their own right." We take Reviewer DJMW's constructive feedback seriously about better framing our narrative around compositional generalization, which we address below along with other valuable concerns raised.

Specifically, we address: (1) the relationship between our SIM task and "out-of-distribution generalization" (Reviewer DJMW, pvo7), (2) the connection between our theoretical analysis and practical diffusion models (Reviewer pvo7, c6mH), and (3) the need for more detailed explanation about the diffusion model experiment results in Section 5 (Reviewer DJMW, pvo7).

---

> ### Author Response · Authors · 2024-11-24
> **General Response (Part 2)**
>
> ## Small variance Gaussian mixtures ensure SIM Task tests compositional (OOD) generalization.
>
> Reviewers DJMW and pvo7 raised a crucial concern regarding whether the SIM Task provides an appropriate framework for studying out-of-distribution (compositional) generalization, noting that with Gaussian training data, "any point in $\mathbb R^d$ has probability$>0$". This valuable feedback highlighted the need to make our key assumptions more explicit, particularly regarding our choices of mean and variance parameters for the Gaussians, which establish our structured Gaussian mixture dataset and test point evaluation as genuinely out-of-distribution. Furthermore, we have **conclusively validated our key results by experimenting with truncated Gaussians**, where we explictly ensured that the probability of sampling training data around the test point is zero. We summarize our key arguments  below:
>
> 1. **Controlled Parameter Regime**: As we visualize in Figure 1\(c\), our study specifically focuses on parameter settings that ensure meaningful out-of-distribution testing:
>    * The training distribution density at test points is exponentially small (technically non-zero but negligible for practical purposes and in our empirics). Concretely, in Figure 1\(c\), we place the Gaussian centroids at the vertices $\{(0,0,0), (\mu_1,0,0),(0,\mu_2,0),(0,0,\mu_3)\}$, while the test point (shown in red) is at $\hat{\mathbf{x}} = (\mu_1,\mu_2,\mu_3)$ -- their vector sum. The visualization clearly demonstrates that the probability density of the training distribution at the test point is practically zero.
>     * We acknowledge that Figure 2\(b\) and the unspecified $\sigma$ in Figure 2\(c\) might have been major sources of confusion. As rightfully pointed out by Reviewer pvo7 and DJMW, selected curves in Figure 2(b) demonstrates extreme cases with extremely high standard deviations ($\sigma=1$ or $2$), where the test point in deed is "in-distribution". However, this setup was specifically chosen to demonstrate an edge case where an extremely large and highly imbalanced $\sigma$ can reverse the direction of the learning dynamics. We emphasize that **all experimental and theoretical results, except for selected curves in Figure 2(b), focus on regimes where the training distribution density at the test point is practically zero**. Therefore, this extremely high $\sigma$ setting should not be regarded as a general configuration of the SIM task. We have incorporated this valuable feedback in our revised manuscript.
>
> 2. **Validation with Strictly OOD Setup**: To definitively address these concerns, we have included a new section (Appendix F.3) in the revision with two new figures:
>     * Visualized in new Figure 15 that the Gaussian clusters constituting training data in Figure 2 \(c\) demonstrating Transient Memorization have negligible overlap in the training data making test point cearly out-of-distribution
>     * Added new experiments using truncated Gaussians that strictly ensure zero training density in the quadrant containing the test point (new Figure 16). Indeed, **our key findings about learning dynamics hold even under these stricter OOD conditions!**
>
> 3. **Clarification on the Theoretical Requirements**: In fact, our theoretical analysis about Transient Memorization inherently requires that $\sigma$ be small relative to $\mu$ (Assumption D.5), so that the test point and training points should be well-separated. We have We have clarified this point in the revision by adding a Remark in Section 4.2.2.
>
> We thank the reviewers again for feedback that enabled us to clarify the key assumptions that were implicit at some places much more explicitly in our manuscript. This setup allows us to study meaningful out-of-distribution generalization while maintaining mathematical tractability. The reproduction of our results with truncated Gaussians firmly confirms that our observations are not artifacts of the theoretical non-zero probabilities in Gaussian tails.
>
> ### **Why is it important to emphasize that Transient Memorization is an OOD phenomenon**
>
> It is worth noting that Transient Memorization implies a non-monotonic test loss curve. It is known that, in the small step size setting, the training loss must monotonically decay. Consequently, if test loss curve is non-monotone, there are typically two reasons:
> 1. The model overfits the training data (over-parameterization);
> 2. The training is noisy (e.g., due to stochastic gradient descent).
>
> However, in this paper, our setting doesn't fall into either of the above categories: we use a small step size, a large amount of data, and full-batch training. Despite these conditions, we still get a non-monotonic test loss curve. As demonstrated by our theoretical analysis, this non-monotonicity is an intrinsic property of the **geometric structure of SIM task itself**. Based on this reason, we think it is crucial to highlight the OOD nature of the SIM task.

---

> ### Author Response · Authors · 2024-11-24
> **General Response (Paer 3)**
>
> ## Substantial Refinements to Section 5: Experimental Results with Diffusion Models
>
> Several reviewers noted that while the paper's overall writing is clear, the Section 5 describing our experimental validation of theoretical predictions using diffusion models needed improvement. This feedback helped us recognize the need for a more careful and thorough introduction to our setup of diffusion models trained on multimodal datasets. In response, we have comprehensively revised Section 5 and enhanced its accompanying figures to improve clarity and comprehension.
>
> ## The SIM Task as a Model for Studying Compositional Generalization in Diffusion Models
>
> Multiple reviewers requested clarification regarding our rationale for using the SIM task as a model for studying compositional generalization in diffusion models. We identified that these concerns stemmed from three main sources:
>
> 1. The need to better explain why our design, using Gaussian mixtures for training and specific test points, constitutes true out-of-distribution (OOD) testing.
> 2. The necessity to revise Section 5 on diffusion model experiments to improve its self-consistency and readability.
> 3. Questions about whether our theoretical framework's simplicity adequately captures the complexity of diffusion models trained on multimodal datasets.
>
> Having addressed the first two points in our previous responses, we now focus on our fundamental argument for why our framework, despite its apparent simplicity, serves as an effective model for studying compositional generalization.
>
> While the definition of a "good model" is a subject of its own, here we adhere to the principle of reducing the complexity of our model setup (i.e. data, task and architecture) as simple as possible, as long as it reproduces a set of phenomena previously characterized in diffusion models.  Our framework has achieved this goal in several ways:
>
> **Our theory reproduces key phenomena previously observed in diffusion models:**
>
> 1. Concept signal determines the order of concept learning.
>     * Demonstrated in Figures 3 and 4 of previous work [1] with diffusion models.
>     * Captured in SIM task as shown in Figure 2 (a) and (b).
>
> 2. Trajectory in concept space show sharp transitions for OOD concept classes, but not for in-distribution classes [1].
>     * Demonstrated  Figure 4 of previous work [1] with diffusion models.
>     * Captured in SIM task as shown in Figure 2\(c\).
>
> 3. Concept distance from the training set governs the order of compositional generalization.
>     * Demonstrated in Figure 5 of previous work [2] with diffusion models.
>     * Captured in SIM task as shown in Figure 7.
>
> **Our theory predicted novel non-monotonic learning dynamics in test loss:**
> * Theoretically predicted by SIM task in Figure 5.
> * Empirically confirmed with diffusion models in Figure 6.
>
> We argue that the SIM task's success in reproducing this rich set of empirical results, despite its simplicity, stems from our focus on abstracting the geometric structure inherent in many compositional generalization scenarios. While we have made certain assumptions to enable theoretical analysis, **we contend that our theory serves as a "productive model" based on two key achievements: (1) providing mechanistic explanations for at least three previously unexplained phenomena, and (2) generating novel predictions that we have validated through diffusion model experiments**.
>
> &nbsp;
>
> [1] Emergence of hidden capabilities: Exploring learning dynamics in concept space
>
> [2] Compositional Abilities Emerge Multiplicatively: Exploring Diffusion Models on a Synthetic Task

---

### Meta-Review · Area_Chair_ZSs8 · 2024-12-15

**Metareview:**

(a) summary

This paper studies how to model the learning dynamics of compositional generalization(CG) in Neural Networks. It introduces a structured identity mapping (SIM) task for analyzing the learning dynamics mathematically. SIM helps to explain the empirical observations in prior work, and it also provides new insights on the non-monotonic learning dynamics in training text-conditioned diffusion models.

(b) strengths
+ It offers novel characteristics of training behavior of basic models.
+ It provides theoretical analysis of dynamic behavior in simple linear models.
+ The empirical results on text-conditioned diffusion models are compelling.
+ It is easy to read and follow.

(c) weaknesses
- The theoretical analysis is on simple linear models, but not on more complex models.
- The studied model is different from practical diffusion models.
- It needs more discussion on the connection between SIM task and CG.
- The details of the experimental setting on SIM was missing.

(d) decision

The reviewers found the paper well-written, the theoretical analysis and empirical results are clearly discussed. The contributions of the paper for bettering understanding of dynamic behavior for achieving CG are solid and relevant, I recommend accept.

**Additional Comments On Reviewer Discussion:**

The reviewers raised some concerns on the relationship between SIM task and "out-of-distribution generalization", the connection between the theoretical analysis and practical diffusion models, and the need for more detailed explanation about the diffusion model experiment results. The authors successfully addressed the concerns in the rebuttal and the reviewers raised their scores to accept.

---

### Decision · Program_Chairs · 2025-01-22

Accept (Poster)